# Evolutionarily conserved long-chain Acyl-CoA synthetases regulate membrane composition and fluidity

**Mario Ruiz[1†], Rakesh Bodhicharla[1†], Marcus Ståhlman[2], Emma Svensk[1], Kiran Busayavalasa[1], Henrik Palmgren[3], Hanna Ruhanen[4,5], Jan Boren[2], Marc Pilon[1]\***

[1]Department of Chemistry and Molecular Biology, University of Gothenburg, Gothenburg, Sweden; [2]Department of Molecular and Clinical Medicine/Wallenberg Laboratory, Institute of Medicine, University of Gothenburg, Gothenburg, Sweden; [3]Metabolism BioScience, Cardiovascular, Renal and Metabolism, BioPharmaceuticals R&D, AstraZeneca, Gothenburg, Sweden; [4]Helsinki University Lipidomics Unit, Helsinki Institute for Life Science, Helsinki, Finland; [5]Molecular and Integrative Biosciences Research Programme, Faculty of Biological and Environmental Sciences, University of Helsinki, Helsinki, Finland

**Abstract** The human AdipoR1 and AdipoR2 proteins, as well as their *C. elegans* homolog PAQR-2, protect against cell membrane rigidification by exogenous saturated fatty acids by regulating phospholipid composition. Here, we show that mutations in the *C. elegans* gene *acs-13* help to suppress the phenotypes of *paqr-2* mutant worms, including their characteristic membrane fluidity defects. *acs-13* encodes a homolog of the human acyl-CoA synthetase ACSL1, and localizes to the mitochondrial membrane where it likely activates long chains fatty acids for import and degradation. Using siRNA combined with lipidomics and membrane fluidity assays (FRAP and Laurdan dye staining) we further show that the human ACSL1 potentiates lipotoxicity by the saturated fatty acid palmitate: silencing ACSL1 protects against the membrane rigidifying effects of palmitate and acts as a suppressor of AdipoR2 knockdown, thus echoing the *C. elegans* findings. We conclude that *acs-13* mutations in *C. elegans* and ACSL1 knockdown in human cells prevent lipotoxicity by promoting increased levels of polyunsaturated fatty acid-containing phospholipids.

**\*For correspondence:**
marc.pilon@cmb.gu.se

[†]These authors contributed equally to this work

## Introduction

Lipotoxicity occurs when fatty acids, especially saturated fatty acids such as palmitate, accumulate at excessive levels in cells or plasma (*Mota et al., 2016*; *Palomer et al., 2018*; *Schaffer, 2016*). In particular, liver steatosis, beta cell failure and endothelial cell defects are caused by lipotoxicity and are important health complications associated with obesity and diabetes. Various factors have been implicated in palmitate-mediated cellular toxicity, including ceramides (*Turpin et al., 2006*), reactive oxygen species (*Gao et al., 2010*), endoplasmic reticulum (ER) stress (*Borradaile et al., 2006*; *Wei et al., 2006*), and small nucleolar RNAs (snoRNAs) (*Michel et al., 2011*). Additionally, recent evidence suggests that a primary mechanism of lipotoxicity relates to membrane rigidification caused by an excess of saturated fatty acids (SFAs) incorporation into membrane phospholipids. For example, adipocytes must promote fatty acid desaturation in order to prevent membrane rigidification and lipotoxicity by palmitate (*Collins et al., 2010*). Also, two recent genome-wide gene silencing/knockout screens identified regulators of SFA incorporation into phospholipids as key determinants of palmitate toxicity in human cells (*Piccolis et al., 2019*; *Zhu et al., 2019*).

Lipotoxicity via membrane rigidification appears evolutionarily conserved. In *C. elegans,* the gene *paqr-2* encodes a homolog of the mammalian AdipoR1 and AdipoR2 (seven transmembrane domain proteins localized to the plasma membrane with their N-terminus within the cytosol and likely acting as hydrolases; *Holland et al., 2011*; *Pei et al., 2011*; *Tanaka et al., 1996*; *Tang et al., 2005*; *Yamauchi et al., 2003*) and acts together with its dedicated partner IGLR-2 (a single-pass plasma membrane protein with a large extracellular domain containing one immunoglobulin domain and several leucine-rich repeats) to sense and respond to membrane rigidification by promoting fatty acid desaturation until membrane fluidity is restored to optimal levels (*Svensson et al., 2011*; *Svensk et al., 2013*; *Svensk et al., 2016a*; *Devkota et al., 2017*; *Bodhicharla et al., 2018*). Wild-type worms are unaffected by the presence of SFAs in their diet, but *paqr-2(tm3410)* or *iglr-2(et34)* null mutants are extremely SFA-sensitive: inclusion of SFAs in the diet of the mutant rapidly leads to excess SFAs in membrane phospholipids, membrane rigidification and death. Both proteins are integral plasma membrane proteins that are also essential for the ability of *C. elegans* to grow at low temperatures such as 15°C because they are required to sense cold-induced rigidification and promote fatty acid desaturation until membrane fluidity is restored (*Svensk et al., 2013*). The *paqr-2 (tm3410)* and *iglr-2(et34)* mutant phenotypes also include a withered appearance of the thin membranous tail tip (*Svensson et al., 2011*; *Svensk et al., 2016b*) and all mutant phenotypes can be attenuated or fully suppressed by secondary mutations in other genes that cause increased fatty acid desaturation (*Svensk et al., 2013*) or increased incorporation of potently fluidizing long-chain polyunsaturated fatty acids (LCPUFAs; fatty acids with 18 carbons or more and two or more double bonds) into phospholipids (*Ruiz et al., 2018*); the *paqr-2/iglr-2* epistatic interaction pathway is summarized in *Figure 1—figure supplement 1*. Additionally, the *paqr-2(tm3410)* and *iglr-2(et34)* mutant phenotypes can be partially suppressed by the inclusion of fluidizing concentrations of nonionic detergents in the culture plate (*Svensk et al., 2013*).

It seems clear that upregulation of desaturases mitigates the membrane-rigidifying effects of SFAs by converting them to more fluidizing monounsaturated fatty acids (MUFAs) and polyunsaturated fatty acids (PUFAs). However, it is much less clear how phospholipid composition can be regulated given a fatty acid pool comprising a mixture of SFAs, MUFAs and PUFAs. In an effort to identify such regulators, we performed a screen in *C. elegans* to identify enhancers of *mdt-15(et14)*, a gain-of-function allele with increased expression of desaturases that partially suppresses the SFA intolerance in *paqr-2(tm3410)* and *iglr-2(et34)* mutants (*Svensk et al., 2013*; *Svensk et al., 2016a*; *Devkota et al., 2017*); *mdt-15* is a homolog of the human mediator subunit MED15 that also regulates genes involved in lipid metabolism (*Yang et al., 2006*). Through this screen, we identified a loss-of-function mutation in *acs-13*, which encodes a long-chain fatty acyl-CoA synthetase, and show that this enzyme localizes to mitochondria where it likely promotes LCFA activation and mitochondrial import. We found that mutation or inhibition of this class of acyl-CoA synthetases in *C. elegans* or human cells protects from SFA lipotoxicity by increasing the relative abundance of LCPUFA-containing phospholipids, which improves membrane fluidity.

## Results

### *acs-13* mutations suppress *paqr-2* mutant defects in an UFA-dependent manner

*paqr-2(tm3410) mdt-15(et14)* double mutants were mutagenized using ethyl methanesulfonate and their F2 progeny screened for the ability to grow into fertile adults within 72 hr when cultivated in the presence of 20 mM glucose, which is converted to SFAs by the dietary bacteria and is therefore an expedient way to provide an SFA-rich diet (*Devkota et al., 2017*). In total 50 000 haploid genomes were screened and six independent mutants were isolated. Of these, four are previously published loss-of-function alleles of the gene *fld-1* (homolog of TLCD1/2 in human); these mutations act independently of *mdt-15(et14)* and cause an increase in the LCPUFA content in phospholipids hence restoring membrane fluidity in *paqr-2* mutants, as previously described (*Ruiz et al., 2018*). A fifth mutant has now been identified as a loss-of-function allele of *acs-13*, which encodes a *C. elegans* sequence homolog of the human long-chain fatty acid acyl-CoA synthetases ACSL1, ACSL5 and ACSL6 that are primarily associated with endoplasmic reticulum (*Young et al., 2018*; *Li et al., 2006*), mitochondria outer membrane (*Young et al., 2018*; *Krammer et al., 2011*; *Lee et al., 2011*;

*Lewin et al., 2001*), and/or peroxisomes (*Islinger et al., 2007*; *Islinger et al., 2010*; *Watkins and Ellis, 2012*). The *acs-13(et54)* allele carries a glycine-to-arginine amino acid substitution at position 125 (G125R), within the proposed N-terminal cytoplasmic domain of the ACS-13 protein, which also contains two predicted transmembrane domains presumed by homology to be embedded in organelles such a mitochondria or ER, and a large cytoplasmic C-terminal domain containing the catalytic domain (see *Figure 1A–B*). That *acs-13(et54)* is a loss-of-function mutation was confirmed in several ways. Firstly, the same G125R mutation and other loss-of-function alleles of *acs-13* were created using CRISPR/Cas9 and found to greatly improve the ability of *paqr-2(tm3410) mdt-15(et14)* double mutants to grow on 20 mM glucose (*Figure 1C–D*; *Figure 1—figure supplement 2*). Secondly, the *acs-13(et54)* and *acs-13(ok2861)*, a deletion allele of *acs-13* obtained from the *C. elegans* Genetics Center, both greatly enhance the ability of *mdt-15(et14)* to suppress the glucose and cold intolerance of the *paqr-2* mutant (*Figure 1E–F*; *Figure 1—figure supplement 3A–B*). *acs-13(et54)* also acts as an enhancer of *cept-1(et10)*, which is another partial *paqr-2(tm3410)* suppressor that acts by promoting fatty acid desaturation, but not as an enhancer of *nhr-49(et8)* which is already a very potent *paqr-2(tm3410)* suppressor (*Svensk et al., 2013*; *Svensk et al., 2016a*), nor of *hacd-1(et12)* which is a relatively poor *paqr-2(tm3410)* suppressor (*Svensk et al., 2013*; *Svensk et al., 2016a*) and does not by itself suppress the glucose intolerance (*Figure 1G*). The ability of *acs-13(et54)* to enhance the effects of both *mdt-15(et14)* and *cept-1(et10)* suggests that it requires the elevated UFA levels characteristic of these mutants (*Svensk et al., 2013*) in order to carry out its *paqr-2 (tm3410)* suppressor function. Indeed, *acs-13(et54)* or *acs-13(ok2861)* are not by themselves effective suppressors of the *paqr-2* mutant glucose and cold intolerance nor of its characteristic withered tail tip defect (*Figure 1E–F*; *Figure 1—figure supplement 3A–C and E–F*). On the other hand, *acs-13 (et54)* can by itself partially suppress the brood size defect of the *paqr-2(tm3410)* mutant, though not as effectively as *mdt-15(et14)*, and does not suppress the defecation rate or life span defects (*Figure 1—figure supplement 4A–C*). Thus *acs-13(et54)* is by itself a weak *paqr-2(tm3410)* suppressor but an enhancer of the UFA-producing *mdt-15(et14)* allele as a *paqr-2(tm3410)* suppressor. Additionally, *acs-13(et54)* enhances the protective effect of the PUFA linoleic acid (18:2) when *paqr-2 (tm3410)* mutant worms are challenged with 2 mM palmitate (a 16:0 SFA), which further indicates a synergistic interaction between PUFA availability and *acs-13(et54)* (*Figure 1—figure supplement 4D*).

## ACS-13 is localized to mitochondria of intestinal and hypodermal cells

Three isoforms of ACS-13 exist that differ at their N-terminus; all three isoforms are affected by the G125R *acs-13(et54)* mutation. Restoring expression of wild-type ACS-13 isoform 'a' using a cDNA transgene driven from the *acs-13* promoter restores glucose intolerance (glucose is here again used as an expedient way to provide an SFA-rich diet since it is converted to SFAs by the dietary *E. coli*; *Devkota et al., 2017*) in *paqr-2(tm3410) mdt-15(et14)*; *acs-13(et54)* triple mutants, which confirms the functionality of the isoform 'a' used in this study (*Figure 2A–C*). The same construct modified to carry GFP fused at the C-terminal end of the ACS-13 protein exhibits GFP-localization specifically on the mitochondria of intestinal and hypodermal cells and co-localizes with Mitotracker Deep Red (*Figure 2D*), but not with the endoplasmic reticulum (ER) marker mCherry::SP12 (*Figure 2—figure supplement 1A*). Detection of ACS-13 protein using Western blotting of cytosol, microsome and mitochondria-enriched subcellular fractions is also consistent with the mitochondrial localization of the ACS-13::GFP protein (*Figure 2—figure supplement 1B*). Expression of *acs-13* was determined by qPCR and found to be unaltered in *paqr-2* mutants but decreased in worms carrying the gain-of-function *mdt-15(et14)* allele (*Figure 2—figure supplement 1C*), suggesting that it may be a negatively regulated downstream target of activated MDT-15. Additionally, the morphology of the mitochondria was often visibly abnormal in *acs-13(et54)* mutants (*Figure 2—figure supplement 1D–E*), and an *acs-13* promoter-driven transcriptional reporter shows strong expression in the intestine and hypodermis (*Figure 2E*). We conclude that ACS-13 is localized to mitochondria in intestine and hypodermis, that its expression is influenced by MDT-15 and that it contributes to maintenance of mitochondria morphology.

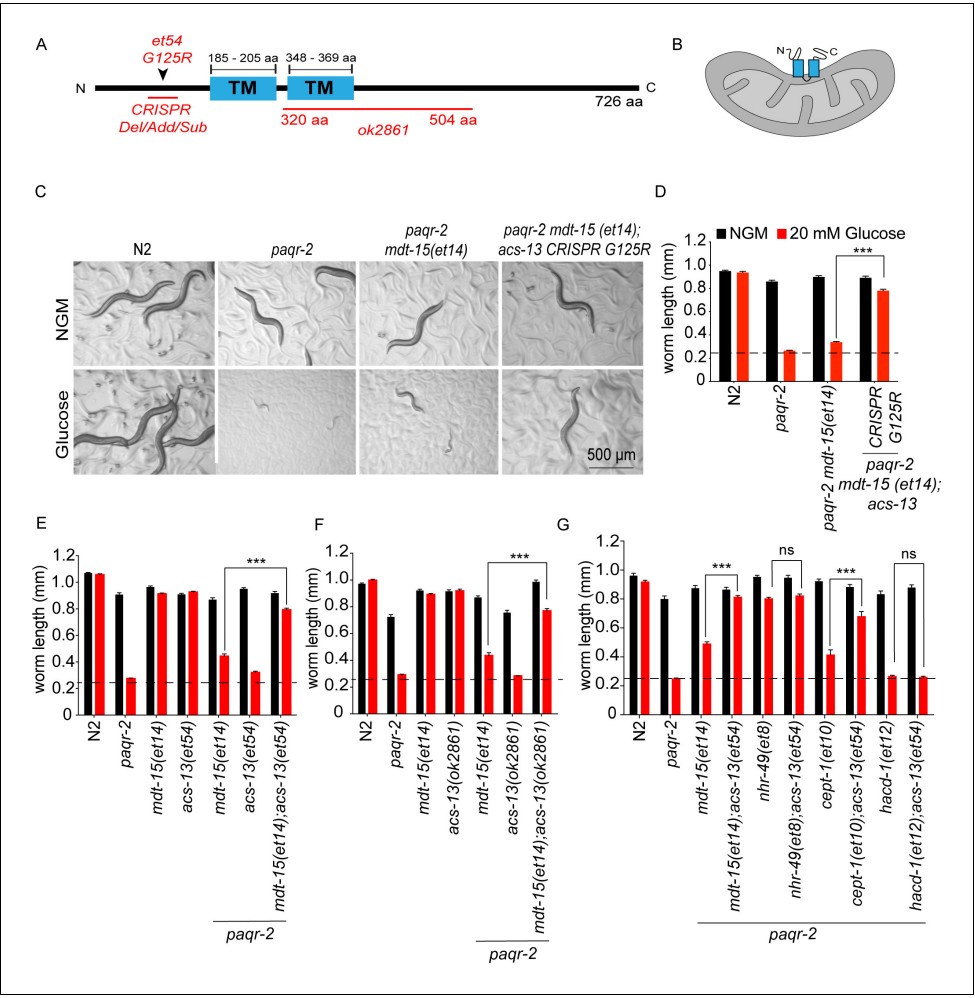

**Figure 1.** *acs-13* mutations enhance the ability of *mdt-15(et14)* or *cept-1(et10)* to suppress the glucose intolerance of the *paqr-2* mutant. (**A**) Cartoon representation of the ACS-13 protein, with the two transmembrane domains indicated in blue and the nature of the amino acid substitution mutant alleles *et54* and the deletion mutant allele *ok2861* indicated in red; the Phyre2 web portal was used for protein modelling and definition of the transmembrane domains (**Kelley et al., 2015**). (**B**) Tentative orientation of ACS-13 inserted in an outer mitochondrial membrane, where the human homolog ACSL1 has been detected (**Lee et al., 2011**; **Lewin et al., 2001**). (**C–G**) Photographs and length measurements of worms with the indicated genotypes placed as L1s on the indicated media (control NGM or 20 mM glucose) then grown for 72 hr (n = 20). The dashed lines in D-G indicate the approximate length of L1s at the start of the experiments. Only the statistical significances of interest are indicated, where ***: p<0.001.

The online version of this article includes the following figure supplement(s) for figure 1:

**Figure supplement 1.** Overview of the likely epigenetic interactions in the *paqr-2* pathway based on published work prior to the present study.

**Figure supplement 2.** Additional CRISPR-generated *acs-13* mutants also enhance the ability of *mdt-15(et14)* to suppress the glucose intolerance in *paqr-2* mutants.

**Figure supplement 3.** *acs-13* mutations enhance the ability of *mdt-15(et14)* to suppress the cold sensitivity of the *paqr-2* mutant.

**Figure supplement 4.** The *acs-13* mutation suppresses the brood size defect in *paqr-2* mutants and enhances the effects of exogenous linoleic acid (18:2 n-6).

## Other Acyl-CoA synthetases can act as *mdt-15(et14)* enhancers

There exist more than twenty acyl-CoA synthetases in *C. elegans* that can activate free fatty acids into acyl-CoAs in preparation for downstream processes such as conjugation into glycerolipids (e.g. phosphatidylcholines [PCs], phosphatidylethanolamines [PEs] or triacylglycerides [TAGs]) or energy-

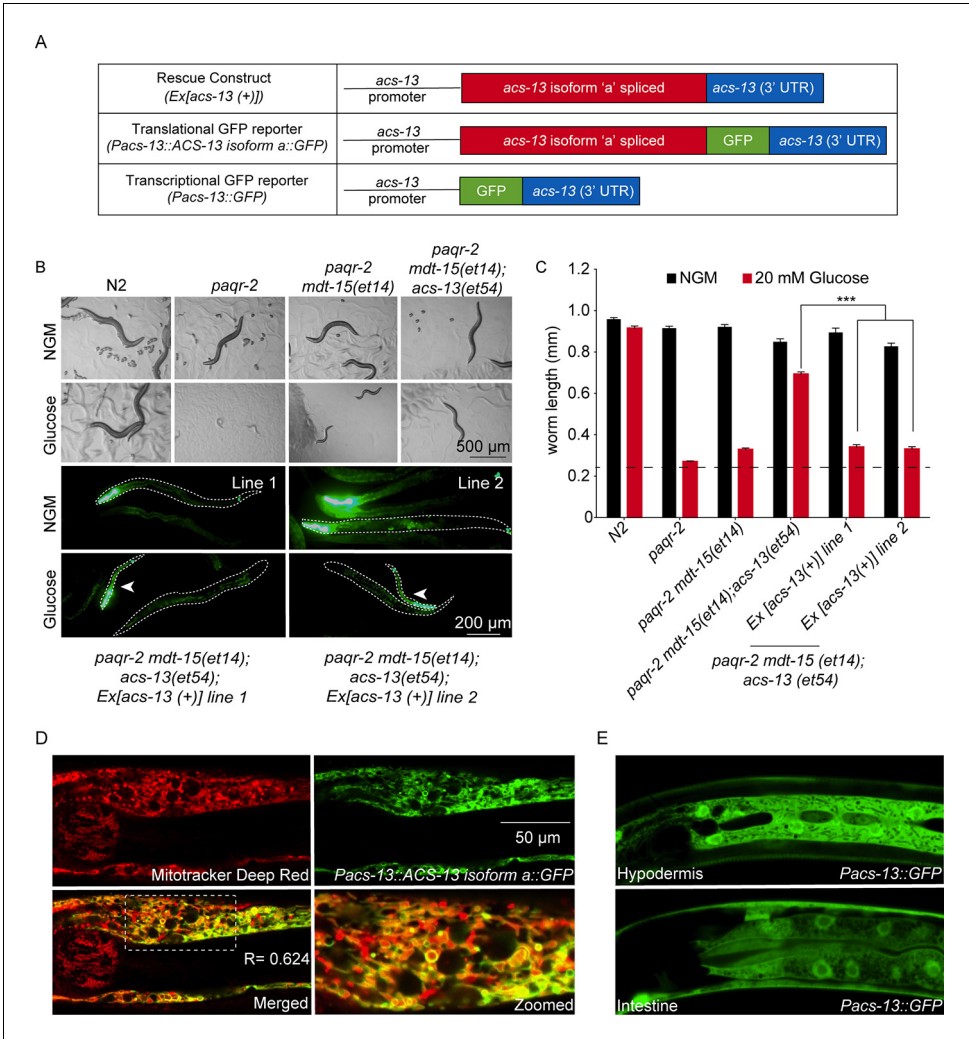

**Figure 2.** The ACS-13 protein localizes to mitochondria and is expressed in intestine and hypodermis. (**A**) Cartoon representation of the *acs-13* constructs created for this study. (**B**) Bright field photographs of worms with the indicated genotypes cultivated on control plates (NGM) or plates containing 20 mM glucose (top two rows), and epifluorescence images of two independent lines of transgenic *paqr-2 mdt-15(et14)* worms carrying the *myo-2:: GFP* marker as well as the wild-type *acs-13(+)* rescue construct and cultivated either on NGM or 20 mM glucose plates (bottom two rows; several worms are outlined with dashed lines). Note how presence of the *acs-13(+)* construct in both transgenic lines abolishes the glucose resistance of *paqr-2 mdt-15(et14); acs-13(et54)* double mutants (examples indicated by arrowheads). (**C**) Length measurements of worms cultivated as in B (n = 20). Only the statistical significances of interest are indicated, where ***: p<0.001. (**D**) Confocal image of the anterior portion of an N2 worms at the L1 stage stained with Mitotracker Deep Red (red) and carrying the *Pacs-13::ACS-13 isoform a::GFP* translational reporter (green). Co-localization is indicated by yellow hues in the merged and zoomed panels. Note the strong colocalization on the periphery of mitochondria, indicating that ACS-13 localizes to the mitochondrial membrane. R is the Pearson correlation coefficient between the two fluorophores. (**E**) Confocal image of N2 worms of the L1 stage and carrying the *Pacs-13::GFP* transcriptional reporter expressed in hypodermis and intestine.

The online version of this article includes the following figure supplement(s) for figure 2:

**Figure supplement 1.** ACS-13 is enriched on mitochondria and important for their morphology.

producing degradation by mitochondria and peroxisomes. We used *E. coli* clones from the Ahringer RNAi library (*Kamath et al., 2003*) to test whether inhibition of 16 different acyl-CoA synthetases (available in the RNAi library) could enhance the ability of *mdt-15(et14)* to suppress the *paqr-2 (tm3410)* mutant glucose intolerance. The most potent *mdt-15(et14)* enhancers were the ACSL1/5/6

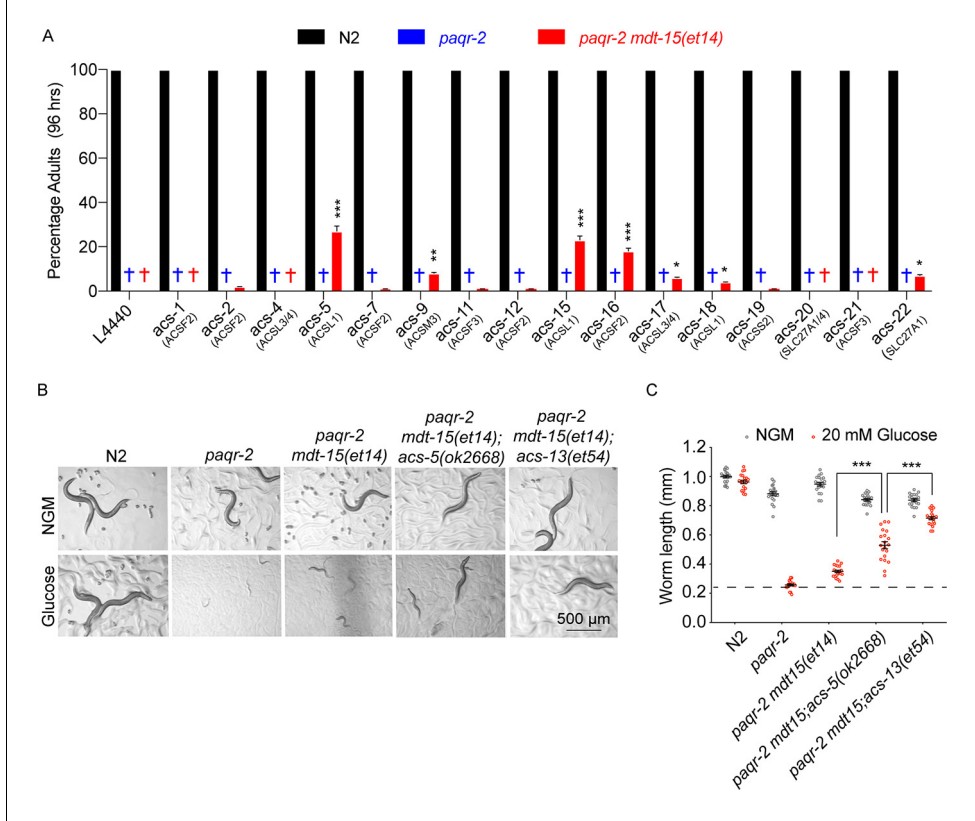

**Figure 3.** Knockdown of several Acyl Co-A synthetases can enhance the ability of *mdt-15(et14)* to suppress the glucose sensitivity of the *paqr-2* mutant. (**A**) Percentage of L1 larvae that grew into adults within 96 hr on plates containing 20 mM glucose (n = 100) and seeded with the indicated RNAi *E. coli* clone (suggested human homolog as per WormBase is indicated in parenthesis; L4440 is the empty vector control). Statistical significances between the empty vector control and RNAi are indicated, where *: p<0.05, **: p<0.01 and ***: p<0.001. (**B–C**) Photographs and length measurements of worms placed as L1s on the indicated media (control NGM or 20 mM glucose) then grown for 72 hr (n = 20). The dashed lines in C indicate the approximate length of L1s at the start of the experiments. Only the statistical significances of interest is indicated, where ***: p<0.001.

homologs *acs-5* and *acs-15*, followed by the ACSF2 homolog *acs-16* (*Figure 3A*). Being homologous to ACSL1/5/6, *acs-5* and *acs-15* may be expected to have preference for LCFA substrates, many of which are PUFAs, and promote their import into mitochondria and/or peroxisomes (*Young et al., 2018*; *Krammer et al., 2011*; *Lee et al., 2011*; *Lewin et al., 2001*; *Islinger et al., 2007*; *Watkins and Ellis, 2012*; *Soupene and Kuypers, 2008*). We obtained the loss-of-function *acs-5 (ok2668)* allele from the *C. elegans* Genetics Center and found that it is indeed an enhancer of *mdt-15(et14)* (*Figure 3B–C*). Like *acs-13(et54)*, *acs-5(ok2668)* did not by itself act as a *paqr-2(tm3410)* suppressor (*Figure 1—figure supplement 3A,D and G*) and is not as potent an *mdt-15(et14)* enhancer as *acs-13(et54)* in suppressing the glucose intolerance (*Figure 3B–C*), which likely explains why only the latter was isolated in our forward genetics screen for *mdt-15(et14)* enhancers.

## The *acs-13(et54)* allele requires desaturase activity but does not cause their upregulation

The fact that *acs-13(et54)* is an enhancer of *mdt-15(et14)* suggests that these two mutations act in separate yet complementary pathways to suppress *paqr-2* mutant phenotypes. Specifically, we hypothesized that *acs-13(et54)* does not act by promoting the expression of fatty acid desaturases, which is the mechanism of action for *mdt-15(et14)*. This is indeed the case: expression of a GFP translational reporter for the Δ9 desaturase FAT-7 is markedly increased in the presence of the *mdt-15(et14)* allele but unaffected by the *acs-13(et54)* allele either by itself or in the *paqr-2(tm3410)* or

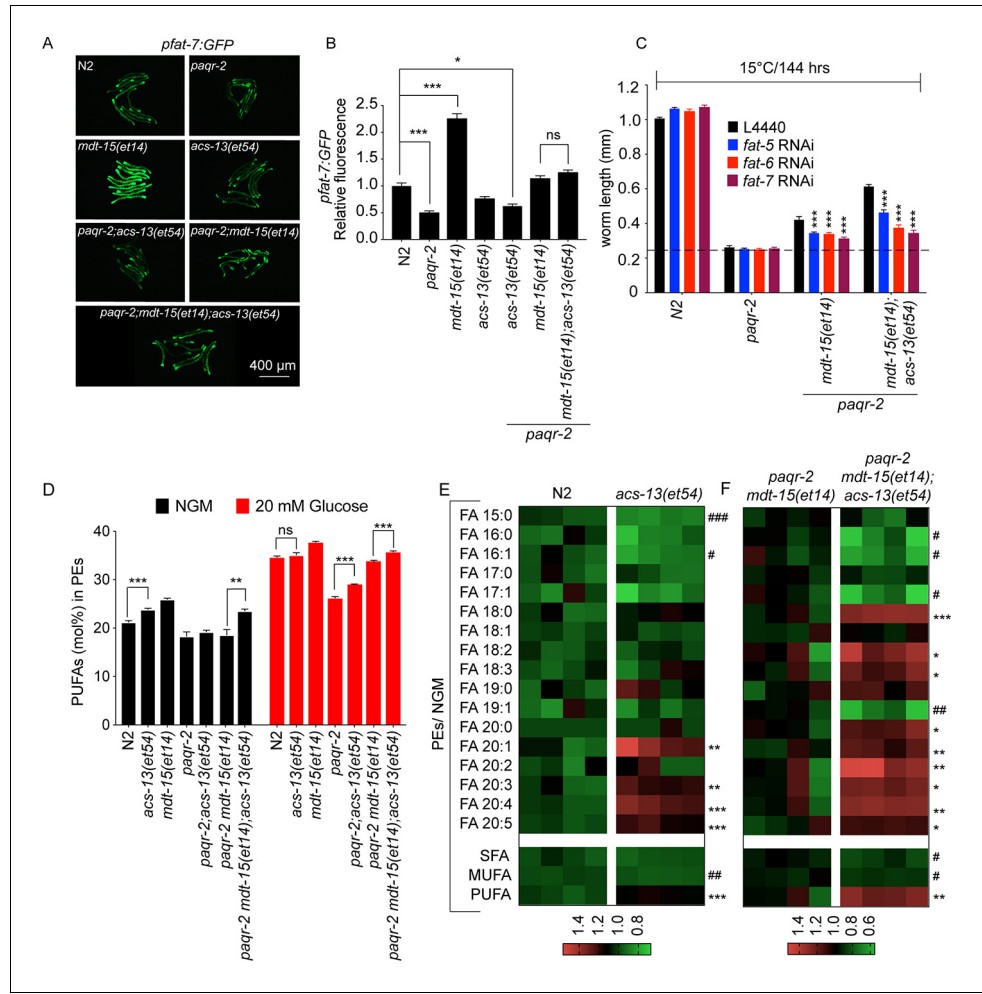

**Figure 4.** The *acs-13* mutation causes elevated LCPUFAs in phospholipids. (**A–B**) *mdt-15(et14)* but not *acs-13(et54)* cause upregulation of the *pfat-7::GFP* reporter (n = 20) on NGM plates. (**C**) RNAi against the Δ9 desaturases *fat-5, fat-6* and *fat-7* reduces the ability of *mdt-15(et14)* and of *mdt-15(et14)* together with *acs-13(et54)* to suppress the cold sensitivity of *paqr-2* mutants (n = 20). (**D**) The *paqr-2* mutant has decreased PUFA levels among PEs on control plates (NGM) or plates containing 20 mM glucose, while the *acs-13(et54)* and *mdt-15(et14)* mutations cause increased PUFA levels among PEs in wild-type worms grown on NGM and together restore normal PUFA levels in *paqr-2* mutants cultivated on NGM plates or 20 mM glucose. (**E–F**) Heat maps showing the relative abundance of specific FA species among the PEs of worms with the indicated genotype and grown on NGM plates. Note the increased abundance of LCPUFAs in genotypes carrying the *acs-13(et54)* mutation. Statistical significances of interest are indicated, where *: p<0.05, **: p<0.01 and ***: p<0.001 for FAs with increased levels in the *acs-13(et54)* mutants (#, ## and ### are used when *acs-13(et54)* mutants have reduced levels of a FA). The online version of this article includes the following source data and figure supplement(s) for figure 4:

**Source data 1.** Lipidomics data for panel D.
**Source data 2.** Lipidomics data for panel E.
**Figure supplement 1.** Enlarged view of *pfat-7::GFP* expression from *Figure 4A*.
**Figure supplement 2.** Lipidomics heat map for FAs among PCs from worms grown on NGM plates or PEs from worms grown on 20 mM glucose.
**Figure supplement 2—source data 1.** Lipidomics data for panel A.
**Figure supplement 2—source data 2.** Lipidomics data for panel B.
**Figure supplement 2—source data 3.** Lipidomics data for panel C.
**Figure supplement 2—source data 4.** Lipidomics data for panel D.
**Figure supplement 2—source data 5.** Lipidomics data for panel E.
**Figure supplement 2—source data 6.** Lipidomics data for panel F.

*Figure 4 continued on next page*

*Figure 4 continued*

**Figure supplement 3.** The *acs-13* mutation enhances the effect of *mdt-15(et14)* on phospholipid FA composition in the *paqr-2* mutant.
**Figure supplement 3—source data 7.** Lipidomics data for panel A.
**Figure supplement 3—source data 8.** Lipidomics data for panel B.
**Figure supplement 3—source data 9.** Lipidomics data for panel C.
**Figure supplement 3—source data 10.** Lipidomics data for panel D.
**Figure supplement 3—source data 11.** Lipidomics data for panel E.

---

*paqr-2(et54) mdt-15(et14)* mutant backgrounds (*Figure 4A–B*; a larger version of *Figure 4A* is presented in *Figure 4—figure supplement 1*). However, the ability of acs-13(et54) to act as an enhancer of mdt-15(et14) is dependent on the activity of desaturases since their inhibition by RNAi significantly impairs the 15°C growth of *paqr-2 mdt-15(et14); acs-13(et54)* triple mutants, with *fat-6* and *fat-7* being particularly important (*Figure 4C*).

## The *acs-13(et54)* mutation causes increased PUFA-containing phospholipids

The human homologs of *acs-13*, such as ACSL1, can activate LCFAs prior to their import into mitochondria (*Soupene and Kuypers, 2008*; *Grevengoed et al., 2015a*; *Coleman, 2019*). A plausible hypothesis regarding the mechanism of action of *acs-13(et54)* is therefore that fewer/different LCFAs (many of which are PUFAs) are being channelled into the mitochondria in mutant worms, and that their accumulation in the cytoplasm leads to their increased activation by other acyl-CoA synthetases that can channel incorporation into phospholipids, resulting in increased membrane fluidity and suppression of *paqr-2(tm3410)* mutant phenotypes. We tested this hypothesis by analysing the fatty acid composition of PEs and PCs in worms carrying different combinations of the *paqr-2(tm3410)*, *mdt-15(et14)* and *acs-13(et54)* alleles. Compared to control N2 worms, we found that the *acs-13(et54)* single mutant has significantly increased levels of PUFAs in the PEs, which are the most abundant membrane phospholipids in *C. elegans*, when worms are grown on NGM (*Figure 4D*). The *mdt-15(et14)* single mutant had increased PUFAs in PEs when grown on NGM plates or plates containing 20 mM glucose (which is converted to SFAs by the dietary *E. coli* [*Devkota et al., 2017*]), as previously described (*Figure 4D*; *Svensk et al., 2016a*; *Devkota et al., 2017*). *paqr-2(tm3410)* mutants had low levels of PUFAs in PEs when cultivated on NGM and much lower levels of PUFAs than control N2 worms when grown in 20 mM glucose, also as previously described (*Figure 4D*; *Svensk et al., 2016a*; *Devkota et al., 2017*). Importantly, *acs-13(et54)* and *mdt-15(et14)* were together better at completely suppressing the low PUFA defects of the *paqr-2* mutant both on NGM and glucose-containing plates (*Figure 4D*). Examining more specifically each type of fatty acid in PEs, we found that presence of the *acs-13(et54)* allele by itself or in the *paqr-2 mdt-15(et14)* background caused a marked increase in LCPUFAs (e.g. 20:3, 20:4 and 20:5) at the expense of shorter SFAs or MUFAs (e.g. 15:0, 16:0 and 16:1) on NGM plates (*Figure 4E*). Similar findings were made with the less abundant PCs and also to a lesser degree on glucose-containing plates (*Figure 4—figure supplements 2* and *3*). We conclude that *acs-13(et54)* acts as a *paqr-2(tm3410)* suppressor because it promotes an increased abundance of PUFA-containing membrane phospholipids, which is a mechanism by which to restore membrane fluidity in the *paqr-2* mutant (*Ruiz et al., 2018*).

## ACSL1 knockdown promotes membrane fluidity in the presence of palmitate in human cells

Sequence comparisons identify the long-chain fatty acyl-CoA synthetases ACSL1, ACSL5 and ACSL6 as the human proteins most similar to the worm *acs-13(et54)*. Based on qPCR, ACSL1 is at least 20-fold more highly expressed than ACSL5 and ACSL6 in HEK293 cells, and all three genes can be successfully knocked down using siRNA (*Figure 5A–B*). The *C. elegans* studies of *acs-13(et54)* point to an important role for this gene in regulating phospholipid composition that ought to impact membrane fluidity. We tested this directly in human cells using the Fluorescence Recovery After Photobleaching (FRAP) method, which relies on the lateral mobility of a membrane associated dye to diffuse laterally and repopulate an area bleached by a laser; the rate of fluorescence recovery in the

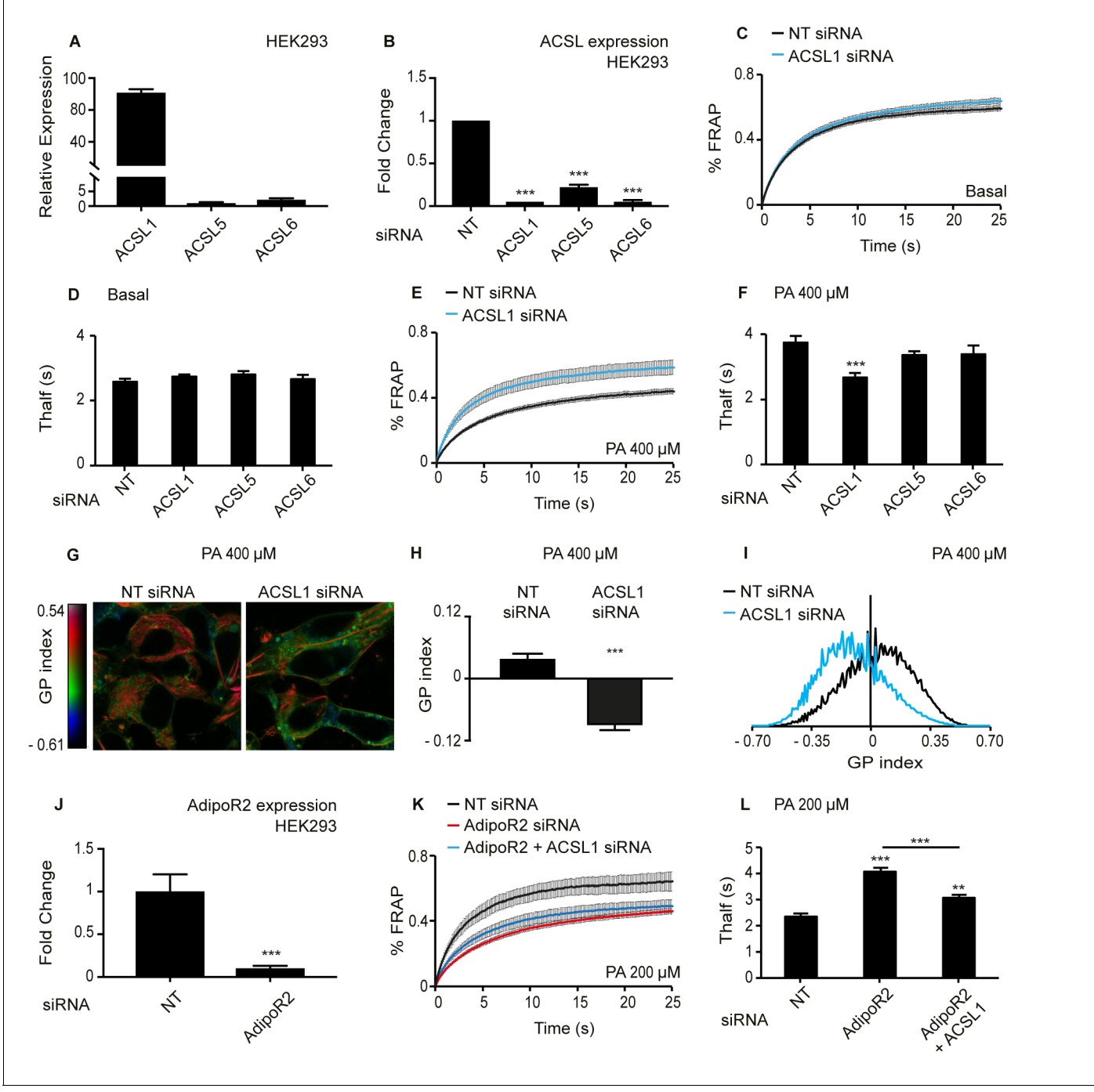

**Figure 5.** Knockdown of ACSL1 protects human HEK293 cells against the membrane-rigidifying effects of palmitate/AdipoR2 knockdown. (A) Relative expression levels of ACSL1, ACSL5 and ACSL6 in HEK293 cells as determined using qPCR. (B) Efficiency of ACSL1, ACSL5 and ACSL6 knockdown using siRNA relative to a non-target (NT) siRNA treatment in HEK293 cells. (C) FRAP analysis showing that siRNA against ACSL1 does not cause a change in membrane fluidity of HEK293 cells under basal conditions compared to NT siRNA treatment. (D) $T_{half}$ values from FRAP experiments with cells treated with the indicated siRNA and cultivated under basal conditions. None of the ACSL siRNA treatments differed significantly from the control NT siRNA; $n \geq 6$. (E–F) As in C-D but in the presence of 400 µM palmitate (PA). Note that ACSL1 siRNA prevented membrane rigidification by palmitate, as evidenced by the significantly lowered $T_{half}$; $n \geq 10$ in F. (G) Pseudocolor images showing Laurdan dye global polarization (GP) index at each pixel position in HEK293 cells treated with NT or ACSL1 siRNA and cultivated in the presence of 400 µM palmitate (PA) for 24 hr. (H) Average GP index from several images as in G; $n = 15$. (I) Distribution of GP index values in representative images for each treatment under basal condition. (J) Efficiency of AdipoR2 knockdown using siRNA relative to a control NT siRNA treatment in HEK293 cells. (K) FRAP analysis showing that AdipoR2 siRNA causes membrane rigidification in HEK293 cells cultivated in the presence of 200 µM PA and that ACSL1 siRNA prevents this rigidification. (L) $T_{half}$ values from FRAP experiments as in K; $n \geq 13$. Statistically significant differences from control are indicated, where **: $p < 0.01$ and ***: $p < 0.001$.

*Figure 5 continued on next page*

*Figure 5 continued*

The online version of this article includes the following figure supplement(s) for figure 5:

**Figure supplement 1.** Silencing either ACSL5 or ACSL6 has no effect on the membrane fluidity of HEK293 cells.

**Figure supplement 2.** Effect of ACSL1 silencing on UPR activation, viability, and expression of other ACSLs.

bleached area reflects membrane fluidity. Under basal conditions, silencing of ACSL1, −5 or −6 has no effect on membrane fluidity (*Figure 5C–D* and *Figure 5—figure supplement 1A*). However, silencing ACSL1 prevents membrane rigidification in HEK293 cells challenged with 400 µM palmitate (*Figure 5E–F*); silencing ACSL5 or ACSL6 had no such effect (*Figure 5F* and *Figure 5—figure supplement 1B*). Using the Laurdan dye method, which reports on water penetration into the membrane that usually correlates with membrane packing and fluidity (*Owen et al., 2012*; *Ruiz et al., 2019*), confirms the FRAP results: HEK293 cells challenged with palmitate have improved fluidity across the entire cell (hence also in organellar membranes) when ACSL1 is knocked down (*Figure 5G–I*); there was no difference in basal media where membrane homeostasis is not challenged (*Figure 5—figure supplement 1C–E*).

AdipoR2 silencing increases the sensitivity of HEK293 cells to the rigidifying effects of palmitate (*Devkota et al., 2017*; *Bodhicharla et al., 2018*; *Ruiz et al., 2018*; *Ruiz et al., 2019*), which is analogous to the SFA-sensitivity in *C. elegans* mutants lacking *paqr-2*, that is a worm AdipoR2 homolog (*Devkota et al., 2017*; *Bodhicharla et al., 2018*; *Ruiz et al., 2018*). Here, we found that silencing ACSL1 abrogates the increased palmitate sensitivity of AdipoR2 siRNA-treated HEK293 cells. Specifically, 200 µM palmitate causes membrane rigidification in HEK293 cells where AdipoR2 is silenced but not when both AdipoR2 and ACSL1 are simultaneously silenced (*Figure 5J–L*). In other words, ACSL1 knockdown acts as a suppressor of AdipoR2 knockdown, just as the *acs-13(et54)* loss-of-function allele acts as a *paqr-2(tm3410)* suppressor in *C. elegans*.

The unfolded protein response (UPR) is induced in conditions of membrane homeostasis defects, including excess lipid saturation that cause membrane thickening and activate the UPR regulator Ire1 (*Halbleib et al., 2017*; *Promlek et al., 2011*; *Volmer et al., 2013*). Using qPCR to monitor the expression of UPR response genes, we found that silencing ACSL1 in HEK293 cells lowers their UPR activation when challenged with 400 µM palmitate, though there was no difference in cell viability (*Figure 5—figure supplement 2A–B*). This observation is consistent with ACSL1 silencing resulting in improved membrane fluidity in the palmitate treated cells such that viability is retained without engaging the UPR response. Mechanistically, ACSL1 silencing could lead to compensatory changes in the expression of other ACSLs, possibly resulting in altered channelling of FAs to different downstream pathways and hence affect membrane composition. However, no obvious changes in expression of other ACSLs were observed when ACSL1 is silenced in HEK293 cells challenged with palmitate, though their relative abundance vis-à-vis ACSL1 have strongly increased (*Figure 5—figure supplement 2C–E*). Additionally, it is interesting to note that ACSL1 expression itself was not changed when silencing AdipoR2 or TLCD1/TLCD2 (*Figure 5—figure supplement 2F*); TLCD1/2 are human homologs of *C. elegans fld-1* and have been described as suppressors of the rigidity defect in AdipoR2 knockdown cells (*Ruiz et al., 2018*); this suggests that expression of ACSL1 is not regulated by AdipoR2 or the TLCDs.

## ACSL1 silencing causes changes in lipid composition and mitochondria homeostasis

Lipidomics analysis shows that, as in *C. elegans*, silencing of ACSL1 leads to a dramatic increase in PUFA-containing membrane phospholipids, with strong increases in the abundance of 20:4, 22:5 and 22:6 in both PCs and PEs of HEK293 cells challenged with palmitate (*Figure 6A*), and also slightly increased PUFA levels in lysophosphatidylcholines (*Figure 6B*). The relative abundance of several types of sphingolipid classes, namely ceramides, dihydroceramides and glucosylceramide, were also decreased by ACSL1 siRNA; levels of sphingomyelins and lactosylceramides were unaffected (*Figure 6—figure supplement 1A–E*). No changes in the PUFA levels within TAGs, in PC/PE and cholesterol/PC ratios or in the actual abundance of several phospholipid classes (PCs, diacyl PEs, alkenyl PEs, phosphatidylinositols and phosphatidylserines) were observed in ACSL1 knockdown cells (*Figure 6C–E* and *Figure 6—figure supplement 1F*). Altogether, these results suggest

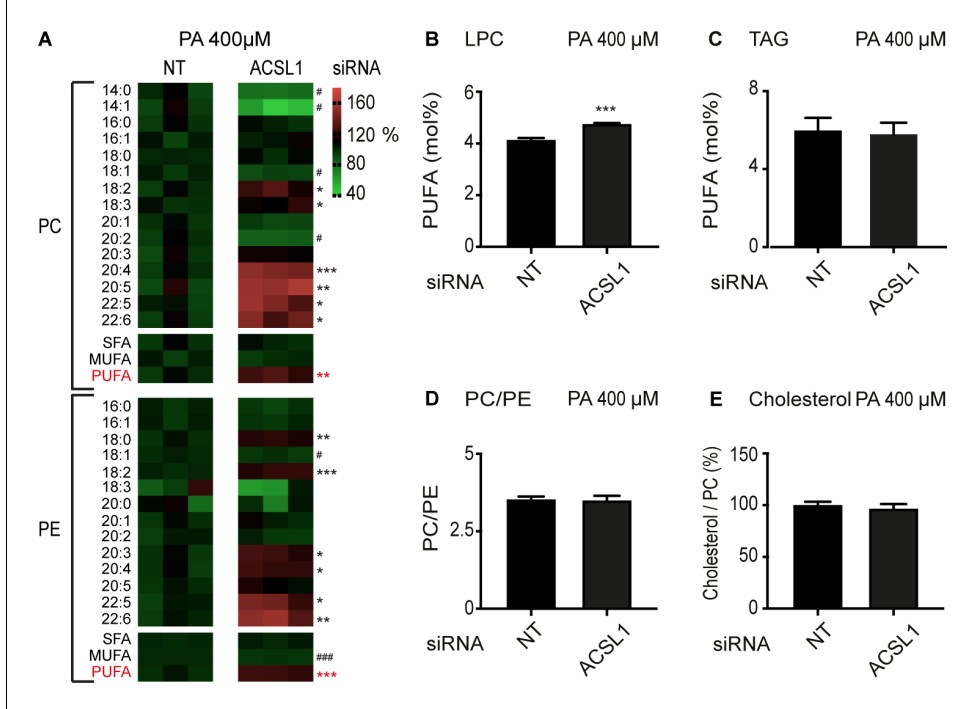

**Figure 6.** ACSL1 siRNA alters specifically membrane phospholipid composition in HEK293 cells challenged with palmitate. (**A**) Heat maps showing the relative abundance of specific FA species among the PCs and PEs of HEK293 cells treated with non-target (NT) or ACSL1 siRNA and cultivated for 24 hr in 400 µM palmitate (PA). Note the increased abundance of PUFA species in the ACSL1 siRNA-treated cells. (**B–C**) Abundance of PUFAs among lysophosphatidylcholines (LPC) and TAGs of HEK293 cells treated with NT or ACSL1 siRNA. Note the increased PUFA levels in the LPCs but not TAGs of the ACSL1 siRNA-treated cells. (**D–E**) PC/PE ratio and abundance of free cholesterol relative to PCs in HEK293 cells treated with NT or ACSL1 siRNA. Statistical significances of interest are indicated, where *: p<0.05, **: p<0.01 and ***: p<0.001 for FAs with increased levels in the ACSL1 siRNA-treated cells (#, ## and ### are used when ACSL1 siRNA-treated cells have reduced levels of a FA).

The online version of this article includes the following source data and figure supplement(s) for figure 6:

**Source data 1.** Lipidomics data for panels A and D.
**Source data 2.** Lipidomics data for panel B.
**Source data 3.** Lipidomics data for panel C.
**Source data 4.** Lipidomics data for panel E.
**Figure supplement 1.** ACSL1 siRNA alters the levels of some sphingolipids in HEK293 cells challenged with palmitate.
**Figure supplement 1—source data 1.** Lipidomics data for panels A to E.
**Figure supplement 1—source data 2.** Lipidomics data for panel F.
**Figure supplement 2.** Effect of ACSL1 siRNA on the lipid composition of HEK293 cells cultivated under basal conditions.
**Figure supplement 2—source data 3.** Lipidomics data for panels A, B and E.
**Figure supplement 2—source data 4.** Lipidomics data for panel C.
**Figure supplement 2—source data 5.** Lipidomics data for panel D.
**Figure supplement 2—source data 6.** Lipidomics data for panel F.
**Figure supplement 2—source data 7.** Lipidomics data for panel G to K.

that silencing ACSL1 in palmitate-challenged cells leads to a specific increase in the levels of PUFAs in several membrane phospholipid classes (PCs and PEs) while leading to a depletion of some sphingolipid classes and no changes in non-membrane lipids (e.g. TAGs). ACSL1 silencing had similar though weaker effects on the lipid composition of cells cultured in basal media (*Figure 6—figure supplement 2*).

The subcellular localization of ACSL1 may vary between cell types (*Soupene and Kuypers, 2008*). ACSL1 has been found in the mitochondria of brown adipocytes (*Ellis et al., 2010*) and cardiac myocytes (*Grevengoed et al., 2015a*), while in hepatocytes it is found on the outer mitochondrial membrane, where it is part of a fatty acid import complex containing also carnitine palmitoyltransferase 1, as well as in the ER and mitochondria associated membranes (*Lee et al., 2011*; *Lewin et al., 2001*). Importantly, ACSL1 can influence the selectivity during LCFA import into mitochondria given its preference for 18:2 and other LCPUFAs (*Grevengoed et al., 2015a*; *Kuwata et al., 2014*). Using subcellular fractionation and Western blotting, we found that ACSL1 is strongly enriched in the mitochondria of HEK293 cells (about 20x enrichment compared to whole lysate; *Figure 7A–B*), just as the ACS-13 protein is associated with mitochondria in *C. elegans*. Some ACSL1 is associated with the ER in HEK293 cells though this appears to be a small minority of the protein (only 3x enrichment in the microsomal fraction; *Figure 7—figure supplement 1A–B*). The predominant association of ACSL1 with mitochondria could be important for channelling LCFAs into that organelle either for beta-oxidation or for mitochondria membrane homeostasis. If that is the case, then ACSL1 knockdown should result in changes in mitochondria composition and/or activity. Consistently, we found a pronounced excess of palmitoylcarnitine in the mitochondria of ACSL1 siRNA-treated cells accompanied by a dramatic reduction in the high-performance liquid chromatography (HPLC) peak containing cardiolipins, which also contained slightly but significantly more 16:0 and 18:2 FA chains and significantly less 18:1 FA chains in the ACSL1 siRNA-treated cells (*Figure 7C–F* and *Figure 7—figure supplement 1C*). Expression of several genes implicated in CL synthesis (CRLS1) and remodelling (Tafazzin and LCLATA1), was reduced in cells treated with ACSL1 siRNA (*Figure 7G*); the downregulation of Tafazzin is in agreement with a previously published study of a mouse ACSL1 knockout model (*Grevengoed et al., 2015a*). The mitochondria lipid composition defect in HEK293 cells treated with ACSL1 siRNA echoes the mitochondria morphology defects observed in *C. elegans*, as one would expect given the important roles played by cardiolipins in regulating mitochondria morphology (*Choi et al., 2006*; *Claypool et al., 2008*; *Kawasaki et al., 1999*; *Steenbergen et al., 2005*). Interestingly, mitochondrial respiration, measured using a pyruvate/glucose rich media to support the citric acid cycle, was not detectably different in HEK293 cells challenged with palmitate and treated with either control or ACSL1 siRNA (*Figure 7H–I* and *Figure 7—figure supplement 1D–H*).

## ACSL1 silencing protects primary human cells against palmitate-induced rigidification

Cancer cell such as HEK293 are characterized by severe abnormalities in many aspects of their metabolism, including lipid metabolism (*Baenke et al., 2013*; *Peck and Schulze, 2016*; *Vriens et al., 2019*). We therefore wished to determine if ACSL1 can also influence membrane fluidity in normal primary human cells. We found that ACSL1 can be effectively silenced using siRNA in human umbilical cord vein endothelial cells (HUVEC) (*Figure 8A*). Using the previously described Laurdan dye method to monitor membrane fluidity (*Owen et al., 2012*; *Ruiz et al., 2019*), we found that silencing of ACSL1 did not affect membrane fluidity in basal conditions (i.e. in the absence of any challenge to membrane homeostasis) but did prevent membrane rigidification in HUVEC cells challenged with 400 µM palmitate (*Figure 8B–G*). ACSL1 knockdown by itself does not overwhelm membrane homeostasis under basal conditions but does protect against the membrane rigidifying effects of palmitate both in HEK293 cells and in human primary endothelial cells.

## Discussion

The present work adds to the mounting evidence that lipotoxicity by SFAs is due to their membrane rigidifying effects: *acs-13* mutations in *C. elegans* and ACSL1 knockdown in human cells prevent SFA toxicity by promoting increased levels of PUFA-containing phospholipids that contribute to maintaining membrane fluidity. Our findings reinforce those of Zhu et al. who performed a CRISPR/Cas9 genome-wide screen to identify modifiers of palmitate toxicity in human cells (*Zhu et al., 2019*). Their top hit for decreasing palmitate toxicity was ACSL3, which acts by activating SFAs such as exogenously provided palmitate; mutating ACSL3 therefore decreases the incorporation rate of SFAs into phospholipids, hence helping to maintain membrane fluidity. Conversely, their top hit for increasing palmitate toxicity was ACSL4, which acts by activating UFAs such that they can be

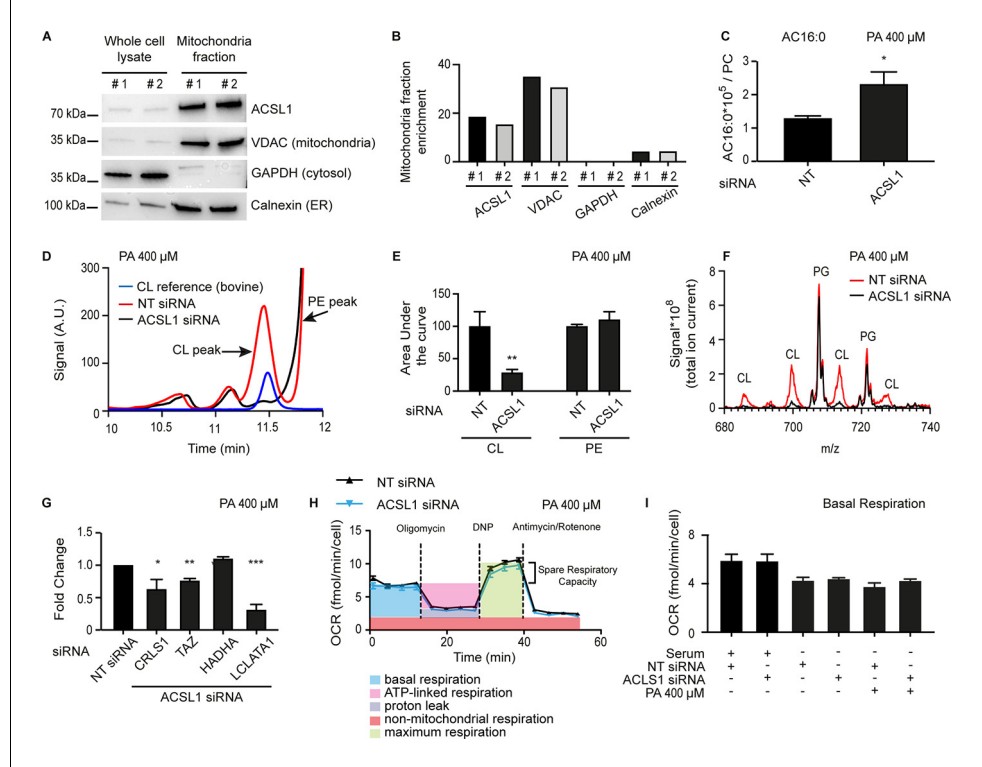

**Figure 7.** ACSL1 is enriched on mitochondria and its deficiency alters acyl-carnitine and cardiolipin levels. (**A**) Western blot showing that ACSL1 is enriched in the mitochondrial fraction purified from HEK293 cells. The VDAC, GAPDH and Calnexin proteins were also detected as markers of mitochondria, cytosol and ER, respectively. (**B**) Quantification of the blot in A. (**C**) Levels of palmitoylcarnitine (AC16:0) relative to total PCs in HEK293 cells treated with non-target (NT) or ACSL1 siRNA and challenged with 400 µM palmitate (PA) for 24 hr. (**D**) HPLC chromatogram for HEK293 cells treated with NT or ACSL1 siRNA. Note that the HPLC peak containing cardiolipins (CLs) is dramatically reduced in the ACSL1 siRNA-treated cells challenged with PA. (**E**) Quantification of the CL and PE peaks from triplicate experiments as in (**D**). (**F**) Mass spectrum confirming that CLs are the dominant species depleted in the peak labelled as 'CL' in ACSL1 siRNA-treated cells in (**D**); levels of phosphatidylglycerols (PGs) were unchanged. (**G**) Changes in the expression of the indicated CL synthesis/remodelling genes in ACSL1 siRNA-treated HEK293 cells challenged with PA. (**H**) Example of a Seahorse analysis measuring oxygen consumption rates in in HEK293 cells treated with NT or ACSL1 siRNA and cultivated in the presence of PA. (**I**) Basal respiration in HEK293 cells pre-treated with the indicated conditions prior to the Seahorse analysis. All PA treatments used 400 µM for 24 hr. Statistically significant differences from control are indicated, where *:p<0.05; **: p<0.01 and ***: p<0.001.

The online version of this article includes the following source data and figure supplement(s) for figure 7:

**Source data 1.** Lipidomics data for panel C.

**Figure supplement 1.** Some ACSL1 is enriched in the microsomal fraction and evidence that mitochondria respiration is still functional when ACSL1 is silenced.

**Figure supplement 1—source data 1.** Lipidomics data for panel C.

---

incorporated into phospholipids; mutating ACSL4 therefore decreases the incorporation rate of UFAs into phospholipids, and thus exacerbates the membrane-rigidifying effects of palmitate. Our work suggests a third mechanism by which ACSLs influence palmitate toxicity: ACS-13 (in *C. elegans*) or ACSL1 (in human) act by activating LCFAs on mitochondrial membranes and promote their import into that organelle where they are utilized for mitochondrial membrane homeostasis or degraded. Mutating or inhibiting ACS-13 or ACSL1 results in more PUFAs available for incorporation into phospholipids, which improves membrane fluidity (see model in *Figure 9*). Other mutations that increase PUFA levels, such as the *mdt-15(et14)* mutation in *C. elegans*, synergize with *acs-13* loss-of-function mutation since they lead to even more PUFAs accumulating and available for incorporation into phospholipids.

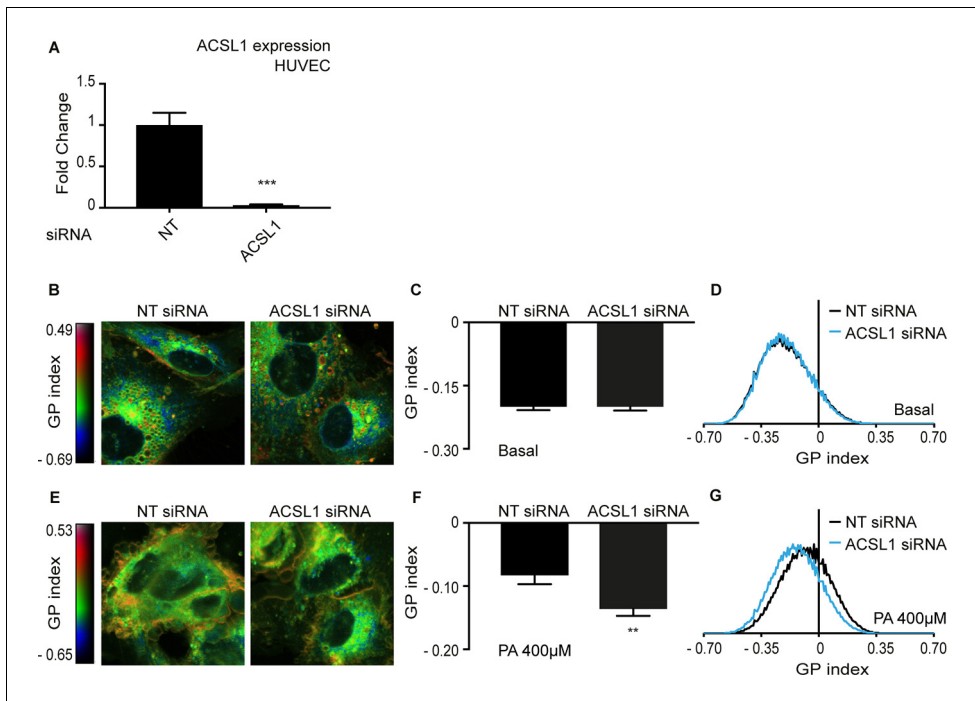

**Figure 8.** Knockdown of ACSL1 protects human primary endothelial cells against palmitate toxicity. (**A**) Efficiency of ACSL1 knockdown using siRNA relative to a non-target (NT) siRNA treatment in HUVEC. (**B**) Pseudocolor images showing Laurdan dye global polarization (GP) index at each pixel position in HUVEC cells treated with NT or ACSL1 siRNA under basal conditions. (**C**) Average GP index from several images as in A; n = 10. (**D**) Distribution of GP index values in representative images for each treatment under basal condition. (**E–G**) As in A-C but with HUVEC cells cultivated in the presence of 400 µM palmitate (PA); n = 15 in E. Statistically significant differences from control are indicated, where **: p<0.01 and ***:p<0.001.

There are 26 acyl-CoA synthetases in human classified into six families based on sequence homology and fatty acid chain length preferences: the ACS short-chain family, ACS medium-chain family, ACS long-chain family, ACS very long-chain family, ACS bubblegum family and the ACSF family) (*Watkins et al., 2007*). Nearly all pathways of fatty acid metabolism require the acyl-CoA synthetase-mediated conversion of free fatty acids to acyl-CoAs. There are five ACSLs (ACSL1 and ACSL3-6) that activate preferentially long-chain fatty acids, and these are intrinsic membrane proteins whose active sites face the cytosol to produce acyl-CoAs that can partition in the proximate membrane monolayer or be transported to different organelles by cytosolic acyl-CoA binding proteins. The subcellular location of each ACSL partly explains how they each direct, or 'channel', their acyl-CoA products to specific downstream pathways (*Coleman, 2019*). For example, ACSL1 is strongly expressed in adipocytes where it has been localized to mitochondria (*Forner et al., 2009*; *Gargiulo et al., 1999*), though subcellular fractionation studies with different cell types have also localized it to the plasma membrane and ER (*Gargiulo et al., 1999*), vesicles (*Sleeman et al., 1998*) and lipid droplets (*Brasaemle et al., 2004*). Adipocyte-specific ACSL1 knockout mice exhibit a severe defect in mitochondria-based long-chain FA degradation, indicating that channeling LCFAs for degradation in mitochondria is an important function of ACSL1 in these cells (*Ellis et al., 2010*). Our present work shows that the *C. elegans* homolog of ACSL1, namely ACS-13, is also localized to mitochondria, suggesting that this may be the ancestral and evolutionarily conserved site of action for this acyl-CoA synthetase. Furthermore, the increased PUFA levels in the phospholipids of the *acs-13* mutant worms are consistent with ACS-13 being required for LCFA utilization in *C. elegans* mitochondria (either for homeostasis of mitochondrial structural lipids or as fuel), just as ACSL1 is required for LCFA degradation in mouse adipocytes.

We identified two major changes in lipid composition when ACSL1 was silenced in HEK293 cells: 1) Increased PUFA levels among PCs and PEs, which make up the bulk of membrane phospholipids in mammalian cells; and 2) dramatic changes in the levels of cardiolipins and palmitoylcarnitine,

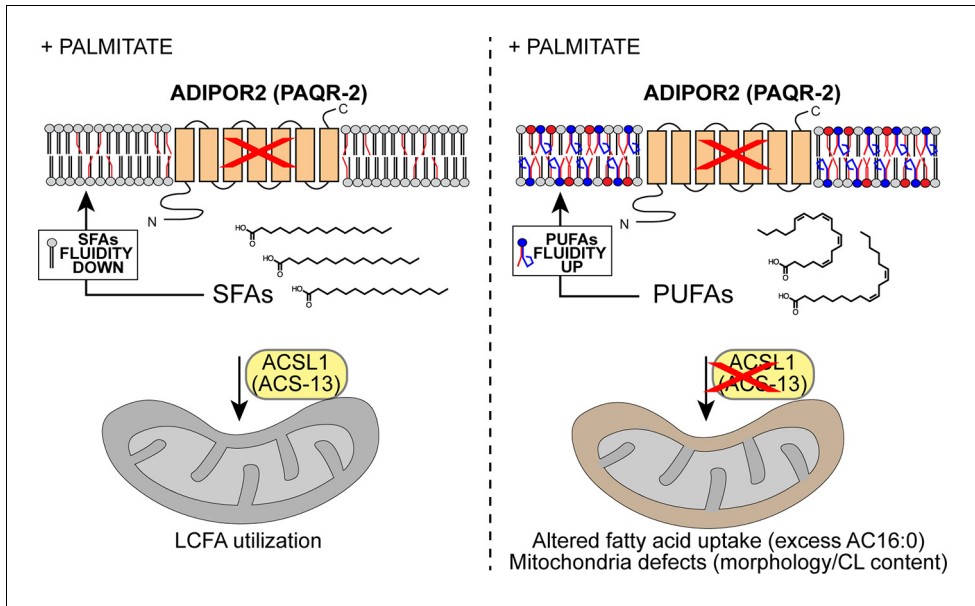

**Figure 9.** Model explaining the membrane-fluidizing effect of decreased/loss of ACSL1/ACS-13. The normal function of ADIPOR2 (PAQR-2 in *C. elegans*) is to monitor and maintain membrane fluidity: it is activated by membrane rigidification and thereby signals to promote fatty acid desaturation and insertion of fluidizing UFAs in phospholipids. When ADIPOR2 is absent, exogenous SFAs become incorporated into the plasma membrane and cause rigidification. When present in mitochondria ACSL1 (ACS-13 in *C. elegans*) help channel LCFAs (including UFAs and PUFAs) into mitochondria. When ACSL1 is absent, fewer/different LCFAs are channelled into mitochondria. This results in mitochondria defects but also increased availability of LCFAs (including UFAs and PUFAs) to be channelled towards other processes, such as phospholipid remodelling that results in improved membrane fluidity.

The online version of this article includes the following figure supplement(s) for figure 9:

**Figure supplement 1.** Revised model of the likely epigenetic interactions in the *paqr-2* pathway based on previously published work and the present study.

which are both mitochondria-specific lipid classes. We speculate that absence of ACSL1 alters the import of LCFAs into mitochondria and that this explains both findings. Grevengoed and co-workers previously drew similar conclusions from a separate study: they found that mouse cardiac myocytes lacking ACSL1 have decreased PUFA levels among mitochondrial phospholipids, including cardiolipins, accompanied by excess PUFAs in PCs and PEs (*Grevengoed et al., 2015a*). To quote a brief passage from their work: "... it is surprising that the linoleate content of PC and PE was also greater in the absence of ACSL. A likely explanation is that when ACSL1 is absent, linoleate increases within the cell and becomes available for activation by other ACSL isoforms that are present on the endoplasmic reticulum where the excess linoleoyl-CoA would be used during the synthesis of PC and PE' (*Grevengoed et al., 2015a*). This is very much in line with the interpretation of our results. Note also that in spite of the mitochondria morphology or lipid composition defects, we observed no growth/brood size phenotypes in the *C. elegans acs-13* mutant nor respiration defects in HEK293 cells treated with ACSL1 siRNA. While there is no doubt that cardiolipins are important for mitochondria morphology and respiration (*Grevengoed et al., 2015a*; *Dudek et al., 2013*; *Li et al., 2010*; *Nguyen et al., 2016*; *Kameoka et al., 2018*), they are not strictly essential, as evidenced by sustained respiration in yeast mutants lacking cardiolipins (*Koshkin and Greenberg, 2000*). Similarly, Barth syndrome patients have severe cardiolipin deficiencies but are viable with cells capable of efficient respiration in spite of their abnormal mitochondria morphology (*Gonzalvez et al., 2013*). These studies therefore suggest that mitochondria are robust and can tolerate the types of defects observed in *C. elegans acs-13* mutants and HEK293 cells where ACSL1 has been silenced. Note also that the mitochondria morphology defects in the *C. elegans acs-13* mutant could reflect either a direct failure in membrane homeostasis due to poor LCFA import, or some indirect consequence

such as impaired autophagy, as has been previously proposed for mouse cardiac myocytes lacking ACSL1 (*Grevengoed et al., 2015b*).

Alternative explanations for some of our observations remain possible. In particular, association of ACSL1 to mitochondria can help tether lipid droplets to the mitochondrial outer membrane, hence facilitating uptake of TAG-released FAs (*Young et al., 2018*; *Coleman, 2019*). Though we did not observe changes in TAG composition when ACSL1 was silenced in HEK293 cells, it is possible that FAs released from lipid droplets were diverted to different fates, including phospholipid remodeling, in these cells. Also, the possible roles of peroxisomes, with which ACSL1 can associate (*Islinger et al., 2007*; *Islinger et al., 2010*; *Watkins and Ellis, 2012*), has not been explored in our study; deficient channeling of LCFAs into peroxisomes in ACSL1-deficient cells could also explain many of our observations.

The present results allow us to re-evaluate the epistatic interactions important for membrane homeostasis in *C. elegans*. Tolerance to dietary SFAs and cold adaptation in *C. elegans* both require the plasma membrane PAQR-2/IGLR-2 complex, which likely acts as a sensor of membrane rigidification that can signal to promote adaptive changes in membrane composition to restore fluidity (*Svensson et al., 2011*; *Svensk et al., 2013*; *Svensk et al., 2016a*; *Devkota et al., 2017*). Several *paqr-2(tm3410)* suppressor mutants have now been identified and fall into two broad classes. The first class of *paqr-2(tm3410)* suppressors are mutations that promote the production of UFAs, such as gain-of-function mutations in *mdt-15* or *nhr-49*, that are homologous to mediator subunit MED15 and the nuclear hormone receptors PPARα/HNF4 respectively, and act as transcriptional activators for the Δ9 desaturases (*Svensk et al., 2013*; *Yang et al., 2006*; *Ratnappan et al., 2014*; *Taubert et al., 2006*; *Lee et al., 2015*), or loss-of-function mutations in enzymes of the PC synthesis pathway, such as *pcyt-1* (homologous to human PCYT1A that regulates the rate-limiting step during PC synthesis) or *cept-1* (a homolog of human choline/ethanolamine phosphotransferase, i.e. CEPT1), that result in SBP-1 (homologous to human SREBPs) activation hence increased transcription of Δ9 desaturases (*Smulan et al., 2016*; *Walker et al., 2011*). The second class of *paqr-2* suppressors are mutations that promote increased incorporation of LCPUFAs in phospholipids, such as mutations in *fld-1* (homologs of TLCD1/2 in humans), which encodes a multi-pass plasma membrane protein that limits the incorporation rate of LCPUFAs into phospholipids and may be a regulator of the Lands cycle (*Ruiz et al., 2018*), and *acs-13*, which is the subject of the present study. Suppressor mutations from either class by themselves are not sufficient to fully suppress the *paqr-2(tm3410)* mutant phenotypes, especially with respect to growth in the presence of dietary SFAs which is a particularly severe challenge for this mutant. However, combining suppressors of both classes provides excellent *paqr-2(tm3410)* suppression, as illustrated by the powerful suppression of the *paqr-2(tm3410)* SFA intolerance when combining the *mdt-15(et14)* and *acs-13(et54)* mutations. This suggests that the complete *paqr-2* downstream program likely includes both the induction of desaturases as well as suppression of processes that tend to deplete LCFAs, such as their oxidation in mitochondria. A revised model of epistatic interactions implicated in membrane fluidity homeostasis is presented in *Figure 9—figure supplement 1*. It will be interesting in the future to leverage protein-protein interaction studies (e.g. co-immunoprecipitation) or transcriptome studies (e.g. RNAseq) to try and identify the precise nature of the *paqr-2* downstream effectors.

In conclusion, we showed that suppressing the activity of ACS-13 (in *C. elegans*) or its homolog ACSL1 (in human cells) prevents SFA-induced membrane rigidification and lipotoxicity by increasing the abundance of LCPUFA-containing phospholipids. Exploiting the differences among the ACSLs may allow the targeting of specific acyl-CoA synthetases, such as ACSL1, with small-molecule inhibitors and thus open new therapeutic avenues against lipotoxicity in clinical contexts such as dyslipidemia or liver steatosis.

## Materials and methods

### Key resources table

| Reagent type (species) or resource | Designation | Source or reference | Identifiers | Additional information |
|---|---|---|---|---|

*Continued on next page*

*Continued*

| Reagent type (species) or resource | Designation | Source or reference | Identifiers | Additional information |
|---|---|---|---|---|
| Strain, strain background (*C. elegans*) | N2 | *C. elegans* Genetics Center (CGC) | | |
| Strain, strain background (*C. elegans*) | HA1842 *[rtIs30(pfat-7::GFP)]* | Gift from Amy Walker; PMID: 22035958 | | |
| Strain, strain background (*C. elegans*) | PD4251 *(ccIs4251 [(pSAK2) myo-3p::GFP::LacZ::NLS + (pSAK4) myo-3p::mitochondrial GFP + dpy-20(+)] I)* | *C. elegans* Genetics Center (CGC); PMID: 9486653 | | |
| Strain, strain background (*C. elegans*) | | | | |
| Strain, strain background (*C. elegans*) | | | | |
| Genetic reagent (*C. elegans*) | *paqr-2(tm3410) mdt-15(et14)* | *C. elegans* Genetics Center (CGC); PMID: 24068966 | QC127 | |
| Genetic reagent (*C. elegans*) | *acs-13(et54)* | This paper | | Will be deposited at CGC. |
| Genetic reagent (*C. elegans*) | *acs-13(ok2861)* | *C. elegans* Genetics Center (CGC); PMID: 23173093 | RB2147 | |
| Genetic reagent (*C. elegans*) | *cept-1(et10)* | *C. elegans* Genetics Center (CGC); PMID: 24068966 | QC123 | |
| Genetic reagent (*C. elegans*) | *paqr-2(tm3410)* | *C. elegans* Genetics Center (CGC); PMID: 21712952 | QC129 | |
| Genetic reagent (*C. elegans*) | *nhr-49(et8)* | *C. elegans* Genetics Center (CGC); PMID: 24068966 | QC121 | |
| Genetic reagent (*C. elegans*) | *hacd-1(et12)* | *C. elegans* Genetics Center (CGC); PMID: 24068966 | QC125 | |
| Genetic reagent (*C. elegans*) | *svIs136 [pVB641OB (Pvha-6::mCh::SP12)]* | Gift from Gautam Kao (Univ Gothenburg) | | |
| Genetic reagent (*E. coli* strain HT115) | *L4440; acs-1; acs-2; acs-4; acs-5; acs-7; acs-9; acs-11; acs-12; acs-15; acs-16; acs-17; acs-18; acs-19; acs-20; acs-21; acs-22* | PMID: 12529635 | | The RNAi clones are available within the 'C. elegans RNAi collection (Ahringer)' distributed by Source Bioscience Limited. |
| Cell line (*Homo sapiens*) | HEK293 | ATCC | CRL-1573 | |
| Biological sample (*Homo sapiens*) | HUVEC | Gibco | C-015–5C | Human umbilical vein endothelial cells |

*Continued on next page*

*Continued*

| Reagent type (species) or resource | Designation | Source or reference | Identifiers | Additional information |
|---|---|---|---|---|
| Antibody | Mouse anti-GFP monoclonal antibody (GF28R) | Invitrogen | MA5-15256 | |
| Antibody | Mouse anti-tubulin monoclonal antibody | Sigma | T5168 | |
| Antibody | Goat HRP-conjugated anti-mouse IgG antibody | Dako | P0447 | |
| Antibody | rabbit anti-ACSL1 monoclonal antibody | Cell Signaling | D2H5 | |
| Antibody | rabbit anti-Calnexin monoclonal antibody | Cell Signaling | C5C9 | |
| Antibody | rabbit anti-GAPDH monoclonal antibody | Cell Signaling | 14C10 | |
| Antibody | rabbit anti-VDAC monoclonal antibody | Cell Signaling | D73D12 | |
| Antibody | HRP-conjugated anti-rabbit IgG antibody | Dako | P0399 | |
| Recombinant DNA reagent | *pPD118.33* | Gift from Andre Fire | RRID:Addgene_1596 | |
| Recombinant DNA reagent | *pPD95.77* | Gift from Andre Fire | RRID:Addgene_1495 | |
| Recombinant DNA reagent | *Pacs-13::acs-13* | This paper | | |
| Recombinant DNA reagent | *Pacs-13::ACS-13 isoform a::GFP* | This paper | | |
| Recombinant DNA reagent | *Pacs-13::GFP* | This paper | | |
| Recombinant DNA reagent | *pRF4 [rol-6(su1006)]* | PMID: 1935914 | | |
| Sequence-based reagent | AdipoR2 siRNA | Dharmacon | J-007801–10 | |
| Sequence-based reagent | NT siRNA | Dharmacon | D-001810–10 | Non-target control |
| Sequence-based reagent | ACSL1 siRNA | Dharmacon | J-011654–06 | |

*Continued on next page*

*Continued*

| Reagent type (species) or resource | Designation | Source or reference | Identifiers | Additional information |
|---|---|---|---|---|
| Sequence-based reagent | ACSL5 siRNA | Dharmacon | J-006327–09 | |
| Sequence-based reagent | ACSL6 siRNA | Dharmacon | J-007748–05 | |
| Peptide, recombinant protein | Recombinant Cas9 protein | Dharmacon | CAS11200 | |
| Peptide, recombinant protein | Bovine Serum Albumin (fatty acid free) | Sigma | A8806 | |
| Peptide, recombinant protein | Bovine Serum Albumin | Sigma | A7906 | |
| Commercial assay or kit | RNeasy Plus Kit | Qiagen | 74134 | |
| Commercial assay or kit | Qproteome mitochondria isolation kit | Qiagen | 37612 | |
| Commercial assay or kit | Pierce BCA Protein Assay Kit | ThermoScientific | 23227 | |
| Commercial assay or kit | RevertAid H Minus First Strand cDNA Synthesis Kit | ThermoScientific | K1631 | |
| Commercial assay or kit | HOT FIREPol EvaGreen qPCR Supermix | Solis Biodyne | 08-36-00001 | |
| Commercial assay or kit | Viromer Blue | Lipocalyx | VB-01LB-01 | |
| Commercial assay or kit | Gibson assembly cloning kit | NEB | E5510S | |
| Commercial assay or kit | Enhanced chemiluminescence detection kit | Millipore | WBKLS0100 | |
| Commercial assay or kit | Lipofectamine RNAiMAX | Invitrogen | 13778100 | |
| Chemical compound, drug | BODIPT 500 /510 C1, C12 | Invitrogen | D3823 | |
| Chemical compound, drug | Laurdan | ThermoFisher | D2350 | |
| Chemical compound, drug | Palmitic acid | Sigma-Aldrich | P0500 | |
| Chemical compound, drug | Linoleic acid acid | Sigma-Aldrich | L1376 | |
| Chemical compound, drug | Viromer Blue | Lipocalyx | | |

*Continued on next page*

*Continued*

| Reagent type (species) or resource | Designation | Source or reference | Identifiers | Additional information |
|---|---|---|---|---|
| Chemical compound, drug | Mitotracker Deep Red FM | Invitrogen | M22426 | |
| Software, algorithm | ImageJ script: nprot.2011.419-S1 | PMID: 22157973 | | |
| Software, algorithm | LipidView software | Sciex | | |
| Software, algorithm | MassHunter software | Agilent Technologies | | |
| Software, algorithm | ZEN software | Zeiss | | |

## *C. elegans* strains and cultivation

The wild-type *C. elegans* reference strain N2 and the mutant alleles studied (except for the novel *acs-13* created in the present study) are available from the *C. elegans* Genetics Center (CGC; MN; USA). The *pfat-7::GFP (rtIs30)* carrying strain HA1842 was a kind gift from Amy Walker (*Walker et al., 2011*), and its quantification was performed as previously described (*Svensk et al., 2013*). The *C. elegans* strains maintenance and experiments were performed at 20°C using the *E. coli* strain OP50 as food source, which was maintained on LB plates kept at 4°C (re-streaked every 6–8 weeks) and single colonies were picked for overnight cultivation at 37°C in LB medium then used to seed NGM plates (*Sulston and Hodgkin, 1988*); new LB plates were streaked every 3–4 months from OP50 stocks kept frozen at −80°C. NGM plates containing 20 mM glucose were prepared using stock solution of 1 M glucose that was filter sterilized then added to cooled NGM after autoclaving.

## Screen for suppressors of SFA intolerance and whole genome sequencing

*paqr-2(tm3410) mdt-15(et14)* double mutant worms were mutagenized for 4 hr by incubation in the presence of 0.05 M ethyl methane sulfonate according to the standard protocol (*Sulston and Hodgkin, 1988*). The worms were then washed and placed on a culture dish. Two hours later, vigorous hermaphrodite L4 animals were transferred to new culture plates. Five days later, F1 progeny were bleached, washed and their eggs allowed to hatch overnight in M9 (22 mM KH2PO4, 42 mM Na2HPO4, 85.5 mM NaCl and 1 mM MgSO4). The resulting L1 larvae were transferred to new plates containing 20 mM glucose then screened 72 hr later for fertile adults, which were picked to new plates for further analysis.

The isolated suppressor alleles were outcrossed 4-to-6 times prior to whole genome sequencing (see below), and 10 times prior to their phenotypic characterization or use in the experiments presented here. The genomes of suppressor mutants were sequenced to a depth of 25-40x as previously described (*Sarin et al., 2008*). Differences between the reference N2 genome and that of the mutants were sorted by criteria such as non-coding substitutions, termination mutations, splice-site mutations, etc. (*Bigelow et al., 2009*). For each suppressor mutant, one or two hot spots, that is small genomic area containing several mutations, were identified, which is in accordance to previous reports (*Zuryn et al., 2010*). Mutations in the hot spot that were still retained after 10 outcrosses were considered candidate suppressors and tested experimentally as described in the text.

## Growth, tail tip scoring and other *C. elegans* assays

For length measurement studies, synchronized L1s were plated onto test plates seeded with *E. coli*, and worms were mounted then photographed 72 hr, 96 or 144 hr later (as indicated). The length of >20 worms was measured using ImageJ (*Schneider et al., 2012*). Quantification of the withered tail tip phenotype was done on synchronous 1 day old adult populations, that is 72 hr post L1 (n ≥ 100) (*Svensk et al., 2013*). Other assays starting with 1 day old adults have also previously been described in details: total brood size (n = 5) (*Svensson et al., 2011*); lifespan (n = 100)

(*Svensson et al., 2011*); defecation period (n = 5; average interval between five defecation was determined for each worm) (*Liu and Thomas, 1994*).

## CRISPR-Cas9 genome editing in *C. elegans*

Guide RNA (gRNA) was designed for *acs-13* as a published protocol (*Paix et al., 2017*). Single guide RNA (sgRNA) was conceived to be amplified using forward and reverse primers containing T7 promoter, Guide RNA and tracrRNA sequences as follows: T7 promoter (5'-taatacgactcactataggg- 3'), Guide RNA (5'-ctctaccagggtgttcgccg- 3') and tracrRNA sequence (5'-gttttagagctagaaatagcaagt-taaaataaggctagtccgttatcaacttgaaaaagtggca ccgagtcggtgcttt- 3'). The forward primer was designed using all 20 nucleotides (nt) of T7 promoter sequence, 20 nt of Guide RNA sequence and the first 20 nt of the tracrRNA sequence. The reverse primer was designed using the last 20 nt of the tracrRNA. Using the forward and reverse primers, sgRNA was amplified using the PUC57-sgRNA expression vector as template (*Shen et al., 2014*). The amplified DNA was then used as template for in vitro transcription (MEGAshortscript T7 high yield transcription kit AM1354 from Promega) to produce the sgRNA (5'-taatacgactcactatagggctctaccagggtgttcgccggttttagagctagaaata gcaagttaaaataaggc-tagtccgttatcaacttgaaaaagtggcaccgagtcggtgctttt-3'). The extracted RNA was purified using MEGA-clear RNA purification kit AM1908 (Promega). Finally, the repair template was designed by introducing the *acs-13(et54)* mutation and 50 nt homology arms on both sides of the mutation (5'-aggctttcctgttcgaagacgcgcgcaccctctaccagggtgttcgccgcAgagcccgtctctc gaacaacgggccgatgctcg-gacgtcgagtcaaaca-3'; the G-to-A mutation is indicated in underlined uppercase).

A sgRNA and repair template for *dpy-10* were included in the microinjection mix to identify successful CRISPR/Cas9 events, as per a published protocol (*Paix et al., 2017*). The final microinjection mix consisted of the following: 3 µl of Cas9 protein (3.2 µg/µl; Dharmacon), 0.5 µl KCl (1M), 0.75 µl HEPES pH 7.4 (1M), 7 µl sgRNA *acs-13*, 1 µl *acs-13(et54)* repair template (10 µM), 7 µl sgRNA *dpy-10*, 1 µl *dpy-10* repair template (10 µM). The injection mix was incubated at 37°C for 15 mins before loading into an injection needle prior to injection into worm gonads. 50 individual Rollers were picked to separate from among the progeny of microinjected worms and allowed to lay eggs. The *acs-13* locus of the F2 progeny was amplified by PCR and sequenced to identify successful *acs-13* gene editing events.

## Construction of plasmids

### *acs-13* rescue construct

The *Pacs-13::acs-13* rescue construct was generated with a Gibson assembly cloning kit (NEB) using the following four DNA fragments: (1) 2 kb upstream regulatory sequence (promoter) of *acs-13* isoform a (amplified using the following primers: 5'-gtgtgtgtgagtgtgtgttttttgctccctccgttttccgt-3' and 5'-gccgtgtctgtgataaccattctgtgtgtttctgtgtttta-3'); (2) isoform a cDNA (amplified using the following primers: 5'-taaacacagaaacacacagaatggttatcacagacacggc-3' and 5'-atcgggggggaacggaatctatggaagtttc-gaatacatcg-3'); (3) *acs-13* 3'UTR (amplified using the following primers: 5'-cgatgtgtattcgaaacttccatagattccgttcccccccgat-3' and 5'-tctattcttttgatttataaggctcaactgacacttttcc-3'); and (4) ampicillin resistance vector *pPD95.77* (amplified using the following primers: 5'-ggaaaagtgtcagtt-gagccttataaatcaaaagaataga-3' and 5'-acggaaaacggagggagcaaaaacacacactcacacac-3'). The assembled plasmid was injected into *paqr-2(tm3410) mdt-15(et14); acs-13(et54)* worms at 20 ng/µl together with 3 ng/µl *pPD118.33* (*Pmyo-2:GFP*; *Davis et al., 2008*), which was used as phenotypic marker used to identify transgenic worms.

### *acs-13* translational GFP reporter

The *Pacs-13::ACS-13 isoform a::GFP* translational GFP reporter was generated with a Gibson assembly cloning kit (NEB) using the following two DNA fragments: (1) The *acs-13* promoter with *acs-13* coding sequence was amplified from *Pacs-13::acs-13* (amplified using the following primers: 5'-tggatgaactatacaaatagattccgttcccccccgattca-3' and 5'-agttcttctcctttactcattggaagtttcgaatacatcg-3'); (2) GFP was amplified from the plasmid *Pfld-1::fld-1::GFP* (*Ruiz et al., 2018*) (amplified using the following primers: 5'-cgatgtgtattcgaaacttccaatgagtaaaggagaagaact-3' and 5'-tgaatcgggggggaacggaatc-tatttgtatagttcatcca-3'). The *Pacs-13::ACS-13 isoform a::GFP* plasmid was injected into N2 worms at 20 ng/µl together with 30 ng/µl *pRF4,* which carries the dominant *rol-6(su1006)* marker used to identify transgenic worms (*Mello et al., 1991*).

### *acs-13* transcriptional GFP reporter

The *Pacs-13::GFP* transcriptional reporter was generated by deleting the *acs-13* gene from *Pacs-13:: ACS-13 isoform a::GFP* using PCR-based mutagenesis (Q5- site directed mutagenesis kit, Biolabs) with the following primers: 5'-atgagtaaaggagaagaactttca-3' and 5'-tctgtgtgtttctgtgtttaagtgg-3'. The *Pacs-13::GFP* plasmid was injected into N2 worms at 20 ng/μl together with 30 ng/μl *pRF4*, which carries the dominant *rol-6(su1006)* marker used to identify transgenic worms (*Mello et al., 1991*).

## Pre-loading of *E. coli* with palmitate or linoleic acid

Linoleic acid (0.324 M stock in ethanol) was diluted in LB media to a final concentration of 0.5 or 0.25 mM. Palmitate (0.1 M stock in ethanol) was diluted in LB media to a final concentration of 2 mM. LB containing fatty acids was then inoculated with OP50 bacteria and shaken overnight at 37°C. The bacteria were then concentrated 10X by centrifugation, and seeded onto NGM plates lacking peptone (200 μl/plate). Synchronized L1 larvae were added to such plates the following day.

## Mitotracker staining and co-localization quantification

Mitotracker Deep Red FM (Invitrogen) was suspended in anhydrous dimethylsulfoxide (DMSO) to a stock solution of 10 mM, which was diluted further in M9 to a working solution of 10 μM. Mitotracker deep red was loaded by incubating the worms for 2 hr with 10 μM of the dye in M9 buffer. These worms were washed several times with M9 buffer and mounted on agarose pads, and then observed and photographed using Zeiss LSM880 confocal microscope. Mitotracker Deep Red FM was excited at 644 nm, and the fluorescence emitted between 657 and 765 nm was collected. The quantification of co-localization was done using ImageJ.

## RNAi in *C. elegans*

All strains were grown on control L4440 RNAi bacteria for one generation at 20°C, then synchronized and L1s placed onto assay RNAi, incubated at 15°C and scored on day 6. Feeding RNAi clones were from the Ahringer RNAi library and were sequenced to confirm their identity, and used as previously described (*Fraser et al., 2000*).

## Imaging and scoring of mitochondria morphology in *C. elegans*

Mitochondrial morphology was analysed in N2 and *acs-13* worms using the strain *PD4251* (*ccIs4251 [(pSAK2) myo-3p::GFP::LacZ::NLS + (pSAK4) myo-3p::mitochondrial GFP + dpy-20(+)] I*) which carries a nuclear and mitochondrial marker transgenes (*Fire et al., 1998*). Synchronized L1 larvae of *myo-3::GFP* and *acs-13; myo-3:GFP* worms were spotted on control plates and incubated at 20°°C. After 72 hr of incubation at 20°C, they were washed off the plate and mounted on agarose pads then observed with a Zeiss LSM880 confocal microscope. Worms were scored based for the presence of clearly abnormal mitochondrial morphology in at least one muscle cell (n ≥ 100 worms). The *svIs136 [pVB641OB (Pvha-6::mCh::SP12)]* was used as an ER marker in some experiments (gift from Gautam Kao; *Billing, 2014*).

## *C. elegans* and HEK293 Shotgun Lipidomics

For worm lipidomics, samples were composed of synchronized L4 larvae (one 9 cm diameter plate/ sample; each treatment/genotype was prepared in five independently grown replicates) grown overnight on OP50-seeded NGM or NGM containing 20 mM glucose. Worms were washed three times with M9, pelleted and stored at −80°C until analysis. For HEK293 lipidomics, cells (prepared in at least three independent replicates) were cultivated in the presence of 400 μM palmitate for 24 hr prior to harvesting using TrypLE Express (Gibco). For lipid extraction, the pellet was sonicated for 10 min in methanol and then extracted according to published methods (*Löfgren et al., 2016*). Internal standards were added during the extraction. Lipid extracts were evaporated and reconstituted in chloroform:methanol [1:2] with 5 mM ammonium acetate. This solution was infused directly (shotgun approach) into a QTRAP 5500 mass spectrometer (Sciex, Toronto, Canada) equipped with a with a TriVersa NanoMate (Advion Bioscience, Ithaca, NY) as described previously (*Jung et al., 2011*). Phospholipids were measured using precursor ion scanning (*Ejsing et al., 2009*; *Ekroos et al., 2003*) and TAGs were measured using neutral loss scanning (*Murphy et al., 2007*). Sphingolipids

from HEK293 cells were measured using ultra performance liquid chromatography coupled to tandem mass spectrometry according to previous publication (*Amrutkar et al., 2015*). Free cholesterol from HEK293 cells was quantified using straight phase HPLC coupled to ELS detection according to previous publication (*Homan and Anderson, 1998*). The data were evaluated using the LipidView software (Sciex, Toronto, Canada). The complete lipid composition data are provided as source data accompanying each lipidomics figure.

## Cardiolipin analysis

Cellular lipids were extracted using the Folch procedure. The total extract was reconstituted in heptane:isopropanol [9:1] and injected onto a 4.6 × 100 mm silica column (Spherisorb, Waters, Milford, MA). Separation was performed according to previous publication (*Homan and Anderson, 1998*). During the analysis, 20% of the sample went for detection using the ELS detector while 80% was fraction collected. The cardiolipin fraction was then evaporated and reconstituted in chloroform: methanol [1:2] with 5 mM ammonium acetate and analyzed using mass spectrometry for determination of cardiolipin species composition. The analysis was made using a QTRAP 5500 mass spectrometer (Sciex, Concord, Canada) equipped with a robotic nanoflow ion source, TriVersa NanoMate (Advion BioSciences, Ithaca, NJ). The detection of cardiolipin species (as $[M-2H]^{2-}$ ions) were made by multiple precursor ion scanning according to previous publication (*Ejsing et al., 2006*).

## ESI-MS/MS of acylcarnitine (AC) and phospholipids

Cellular lipids were extracted according to *Bligh and Dyer (1959)* and dissolved in chloroform/methanol 1:1 (by vol). Immediately before mass spectrometry methanol was added to the samples to give a solution of 1:2 chloroform/methanol, and thereafter 2% NH4OH was added along with an internal lipid standard mixture containing AC12:0 (Sigma-Aldrich) and representatives for all phospholipid classes analyzed. Samples were injected into the electrospray source of a triple quadrupole mass spectrometer (Agilent 6410 Triple Quadrupole; Agilent Technologies) at a flow rate of 10 µl/min, and spectra were recorded using both positive and negative ionization mode. AC was analyzed by MS/MS precursor-ion scanning mode as m/z 85 (P85), and phospholipid species were detected using head-group specific MS/MS scanning modes; phosphatidylcholine as P184, phosphatidylethanolamine (PE) as neutral loss of 141 (NL141), phosphatidylserine as NL87 and phosphatidylinositol as P241. Phospholipid acyl chain assemblies were confirmed using negative mode precursor scans for the acyl fragments released from these lipids or their formate adducts. PE alkenyl-acyl species were confirmed as described previously (*Zemski Berry and Murphy, 2004*) and quantified using MS-scan. Mass spectra were processed by MassHunter software (Agilent Technologies) and individual lipid species were quantified using the internal standards and LIMSA software (*Haimi et al., 2006*).

## Cultivation of HEK293 and HUVEC

HEK293 were grown in DMEM containing glucose 1 g/l, pyruvate and GlutaMAX and supplemented with 10% fetal bovine serum, 1% non-essential amino acids, HEPES 10 mM and 1% penicillin and streptomycin (all from Life Technologies) at 37°C in a water humidified 5% $CO_2$ incubator. Cells were sub-cultured twice a week at 90% confluence. HUVEC (passages 1 to 5) were obtained from Gibco and cultivated as described in *Ruiz et al. (2017)*. Briefly, cells were grown in M200 medium (Gibco) containing the Low Serum Growth Supplement (Gibco) and 1% penicillin and streptomycin. Cells were sub-cultured twice a week at 90% confluence, and cultivated on treated plastic flask and multidish plates (Nunc). For FRAP and Laurdan dye experiments, HEK293 or HUVEC were seeded in glass bottom dishes (Ibidi) pre-coated with 0.1% porcine gelatin (Sigma). The HEK293 cell line was authenticated by genotyping and confirmed to be free of mycoplasma.

## siRNA in HEK293 cells and HUVEC

The following pre-designed siRNAs were purchased from Dharmacon: AdipoR2 J-007801–10, Non-target D-001810–10, ACSL1 J-011654–06, ACSL5 J-006327–09 and ACSL6 J-007748–05. For HEK293 cells, transfection of 25 nM siRNA was performed in complete media using Viromer Blue according to the manufacturer's instructions 1X (Lipocalyx). HUVEC were transfected using 10 nM siRNA and Lipofectamine RNAiMAX Transfection Reagent following the HUVEC optimized protocol

from the manufacturer (Invitrogen). Knockdown gene expression was verified 48 hr after transfection.

## Quantitative PCR (qPCR)

Total *C. elegans* and cellular RNA were isolated using RNeasy Kit according to the manufacturer's instructions (Qiagen) and quantified using a NanoDrop spectrophotometer (ND-1000; Thermo Scientific). cDNA was obtained using a RevertAid H Minus First Strand cDNA Synthesis Kit with random hexamers (Thermo Scientific). qPCR experiments were performed with a CFX Connect thermal cycler (Bio Rad) using Hot FIREpol EvaGreen qPCR SuperMix (Solis Biodyne) and standard primers. The relative expression of each gene was calculated according to the ΔΔCT method (*Livak and Schmittgen, 2001*). Expression of the housekeeping gene PPIA (human samples) and *tba-1* (*C. elegans*) were used to normalize for variations in RNA input. Primers designed for this study were: ACSL1-For (5′-caagcaaacaccacgctgaa-3′), ACSL1-Rev (5′-caccatcagccggactcttc-3′), ACSL3-For (5′-tggatgatagctgcacaggc-3′), ACSL3-Rev (5′-tcggtggctttccatcaaca-3′), ACSL4-For (5′-ttcctccaagtagaccaacgc-3′), ACSL4-Rev (5′-tcggtcccagtccaggtatt-3′), ACSL5-For (5′-gaggccaagacacccttgaa-3′), ACSL5-Rev (5′-attacacgaacccttccgcc-3′), ACSL6-For (5′-gtaccttcaccactcctggc-3′), ACSL6-Rev (5′-gcaggcccagtagttcagtt-3′), CRLS1-For (cacccccagcctgtatgaa-3′), CRLS1-Rev (5-′tggcccagtttcgagcaata-3′), TAZ-For (5-′accaaggagctacactccca-3′), TAZ-Rev (5′-catgtgcctgcctgtgtcta-3′), HADHA-For (5′-ggtttggaggtgaaaccca-3′), HADHA-Rev (5′-caggcggaactggatgtctt-3′), LCLAT1-For (5-′caactctggtgccacaaacg-3′), LCLAT1-Rev (5′-tgagtaggcacattgcaggg-3′), *acs-13*-For (5′-tctactcgaagaatcgcgcc-3′), *acs-13*-Rev (5′-tgggcattg ctccttgaact-3′), *tba-1*-For (tctcgcaggttgtgtcttcc) and *tba-1*-Rev (agcctcatggtaagccttgt). Primer sequences for AdipoR2 and PPIA were previously described (*Ruiz et al., 2019*), as were those for sXBP-1, ATF4, DDiT, HSPA5 (*Oslowski and Urano, 2011*). cDNA samples for *Figure 5—figure supplement 2F* are from a previously published study (*Ruiz et al., 2018*). Changes in gene expression were measured in n ≥ 3 biological independent experiment, including internal technical triplicates.

## HEK293 fatty acid treatment

Palmitate was dissolved in sterile DMSO then mixed with fatty acid-free bovine serum albumin BSA (all from Sigma) in serum-free medium for 15 min at room temperature. The molecular ratio of BSA to PA was 1 to 5.3 when using 400 μM palmitate, and 1 to 2.65 when using 200 μM palmitate. Cells were then cultivated in this serum-free media containing palmitate for 24 hr prior to analysis.

## Fluorescence recovery after photobleaching (FRAP) in HEK293 cells

For FRAP in mammalian cells, HEK293 cells were stained with BODIPY 500/510 $C_1$, $C_{12}$ (4,4-Difluoro-5-Methyl-4-Bora-3a,4a-Diaza-*s*-Indacene-3-Dodecanoic Acid) (Invitrogen) at 2 μg/ml in PBS for 10 min at 37°C (*Devkota et al., 2017*). FRAP images were acquired with an LSM880 confocal microscope equipped with a live cell chamber (set at 37°C and 5% $CO_2$) and ZEN software (Zeiss) with a 40X water immersion objective. Cells were excited with a 488 nm laser and the emission between 493 and 589 nm recorded. Images were acquired with 16 bits image depth and 256 × 256 resolution using a pixel dwell of ~1.34 μs. At least ten (n ≥ 10) pre-bleaching images were collected and then the region of interest was beached with 50% of laser power. The recovery of fluorescence was traced for 25 s. Fluorescence recovery and $T_{half}$ were calculated as previously described (*Devkota et al., 2017*).

## Laurdan dye measurement of membrane fluidity in HUVECs

Live cells were stained with Laurdan dye (6-dodecanoyl-2-dimethylaminonaphthalene) (Thermo Fisher) at 10 μM (HUVEC) or 15 μM (HEK293) for 45 min. Images were acquired with an LSM880 confocal microscope equipped with a live cell chamber (set at 37°C and 5% $CO_2$) and ZEN software (Zeiss) with a 40 × water immersion objective as described before (*Bodhicharla et al., 2018*; *Ruiz et al., 2019*). Cells were excited with a 405 nm laser and the emission recorded between 410 and 461 nm (ordered phase) and between 470 and 530 nm (disordered phase). Pictures were acquired with 16 bits image depth and 1024 × 1024 resolution, using a pixel dwell of ~1.02 μsec. Images were analyzed using ImageJ version 1.47 software (*Schneider et al., 2012*), following published guidelines (*Owen et al., 2012*).

## Mitochondria isolation and ACSL1/ACS-13 Subcellular Localization

HEK293 cells and synchronized adults of various *C. elegans* strains were harvested and washed several times. Mitochondria and microsomal fractions were isolated using the Qproteome mitochondria isolation kit (Qiagen). Protein sample concentrations were quantified using the BCA protein assay kit (Pierce) according to the manufacturer's instructions. Equal amounts of protein were mixed with Laemmli sample loading buffer (Bio-Rad), heated to 37°C for 10 min, and loaded in 4% to 15% gradient precast SDS gels (Bio-Rad). After electrophoresis, the proteins were transferred to nitrocellulose membranes using Trans-Blot Turbo Transfer Packs and a Trans Blot Turbo apparatus with the predefined mixed-MW program (Bio-Rad). Blots were blocked with 5% nonfat dry milk (Bio-Rad) or BSA (Sigma) in PBS (Gibco) containing 0.05% Tween-20 (Sigma) (PBS-T) for 1 hr at room temperature. Blots were incubated overnight at 4°C with primary antibody in the blocking buffer recommended by the antibody supplier. Blots were washed with PBS-T and incubated with the appropriate secondary antibody: swine anti-rabbit IgG/HRP (1:3000; Dako) or goat anti-mouse IgG/HRP (1:5000; Dako) and washed again with PBS-T. Blots were developed with ECL (Immobilon Western; Millipore), and the signal was visualized with a digital camera (VersaDoc; Bio-Rad). PageRuler Plus prestained protein ladder was used to assess molecular weight (Thermo Fisher Scientific). Western blots were quantified by densitometry using Image Lab version six software. Primary antibody used with HEK293 samples: anti-ACSL1 (D2H5, 1:1000), anti-Calnexin (C5C9, 1:1000), anti-VDAC (D73D12, 1:1000), anti-GAPDH (14C10, 1:2500) (all rabbit monoclonal from Cell Signaling). For *C. elegans* samples, the primary antibodies included mouse monoclonal anti-GFP (GF28R, Invitrogen, 1:1000), mouse monoclonal anti-tubulin (T5169, Sigma-Aldrich; 1:1000).

## Trypan blue staining

After 24 hr of treatment, cell supernatant was collected, and cells were detached and mixed again with their respective supernatant. The cell suspension was then mixed 1:1 with a 0.4% trypan blue solution (Gibco) and loaded in a hemacytometer and examined immediately under the microscope. The percentage of positive and negative cells in four quadrants was registered.

## Bioenergetics to assess mitochondrial function

Oxygen consumption rate (OCR) was determined by using a Seahorse XF 96 instrument (Agilent). Cell media composition during the assay was: Seahorse XF base medium (Agilent) supplemented with 5.5 mM glucose, 100 mM pyruvate and 2 mM L-glutamine. The Seahorse XF Cell Mito Stress Test was run using 2 µM oligomycin (Sigma-Aldrich), 200 µM DNP (as uncoupler; Sigma-Aldrich), 1 µM rotenone (Sigma-Aldrich) and 1 µM antimycin A (Sigma-Aldrich) as per the manufacturer's instructions. Cells were stained with both Hoechst 33342 (Thermo Fisher) and propidium iodide (Thermo Fisher) to assess nuclei counts and viability as measured by fluorescence microscopy. OCR values were normalized against viable cells.

## Statistics

Error bars show the standard error of the mean, and *t*-tests were used to identify significant differences between treatments. Asterisks are used in the figures to indicate various degrees of significance, where *: $p < 0.05$; **: $p < 0.01$; and ***: $p < 0.001$. Quantitative experiments were performed at least twice independently (except for the acylcarnitine and cardiolipin lipidomics which were performed once with three independent replicates) and with the number of replicates indicated for each method, meaning that populations of worms or cells were independently grown in separate experiments and analysed.

## Acknowledgements

We acknowledge the Centre for Cellular Imaging at the University of Gothenburg and the National Microscopy Infrastructure, NMI (VR-RFI 2016–00968) for assistance in microscopy. We also thank Ranjan Devkota for help with confocal microscopy and qPCR primer design. Matilda Colm and Lisa Westlund for helping with phenotypic characterization of the *acs-13* mutant. Funding: Cancerfonden, Vetenskaprådet, Carl Tryggers Stiftelse, Diabetesfonden, Swedish Foundation for Strategic

Research, Kungliga Vetenskaps-och Vitterhets-Samhället i Göteborg, Wilhelm och Martina Lundgrens Stiftelse and Tore Nilssons Stiftelse.

## Additional information

### Competing interests
Henrik Palmgren: Affiliated with AstraZeneca. The other authors declare that no competing interests exist.

### Funding

| Funder | Grant reference number | Author |
|---|---|---|
| Vetenskapsrådet | Dnr: 2016-03676 | Marc Pilon |
| Cancerfonden | Dnr 16 0693 | Marc Pilon |
| Carl Tryggers Stiftelse för Vetenskaplig Forskning | CTS 16:365 | Marc Pilon |
| Diabetesfonden | DIA2016-109 | Marc Pilon |

The funders had no role in study design, data collection and interpretation, or the decision to submit the work for publication.

### Author contributions
Mario Ruiz, Rakesh Bodhicharla, Kiran Busayavalasa, Conceptualization, Data curation, Formal analysis, Investigation, Visualization; Marcus Ståhlman, Data curation, Formal analysis, Investigation; Emma Svensk, Conceptualization, Investigation; Henrik Palmgren, Hanna Ruhanen, Investigation, Visualization; Jan Boren, Marc Pilon, Conceptualization, Supervision, Funding acquisition, Project administration

### Author ORCIDs
Marcus Ståhlman (iD) http://orcid.org/0000-0002-4202-0339
Marc Pilon (iD) https://orcid.org/0000-0003-3919-2882

### Decision letter and Author response
Decision letter https://doi.org/10.7554/eLife.47733.sa1
Author response https://doi.org/10.7554/eLife.47733.sa2

## Additional files

### Supplementary files
• Transparent reporting form

### Data availability
All data generated or analysed during this study are included in the manuscript and supporting files. The lipidomics data is provided as a supplementary table.

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
