## [Decision Letter]

Thank you for submitting your article "Evolutionarily conserved long-chain Acyl-CoA synthetases regulate membrane composition and fluidity" for consideration by *eLife*. Your article has been reviewed by three peer reviewers, and the evaluation has been overseen by Tobias Walther as the Reviewing Editor and Vivek Malhotra as the Senior Editor.

The reviewers have discussed the reviews with one another and the Reviewing Editor has drafted this decision to help you prepare a revised submission.

All the reviewers agree that this work is interesting, overall well done and will move the field forward. In particular, the model that a mitochondrial enzyme activates PUFAs and is an essential modulator of oxidation of these fatty acids, and lipotoxiciy is considered very interesting.

However, there are some substantial issues that will need to be addressed before considering this manuscript for publication. Particularly, mitochondrial localization of *acs-13* and its mammalian ortholog needs to be further tested and, data for the mammalian system will need to be strengthened. Most importantly, as multiple reviewers pointed out, the conclusions of substrate preference and involvement of *acs-13* in mitochondrial PUFA oxidation need to strengthened.

In addition, I would like to encourage the authors to edit the manuscript to more carefully consider alternative hypotheses and explanation for their observations.

Specific points raised during the review are:

1) The authors provide no data to test their primary hypothesis that *acs-13* causes increased oxidation of PUFAs. This should be addressed. The preference of mitochondrial acyl-CoA synthetases for PUFAs is crucial to the presented model, but has not been directly addressed. The authors should provide evidence for a selectivity of identified Acyl-CoA synthetases. Alternatively, they should test if Acyl-CoA synthetases with a clearly defined selectivity can phenocopy the functions of the enzymes identified in this study, when targeted to mitochondria.

2) Related to this, the subcellular localization of ACSLs is central to the presented model. The microscopic evidence for a mitochondrial localization, however, is not particularly strong. The authors should validate the subcellular localization by alternative means.

Multiple reviewers noted that the mammalian data need to be strengthened. Related to this, the membrane fluidity assays are poorly explained in this paper. The localization of the mammalian ACSL1 could be studied by complementary methods: biochemical isolation/enrichment of mitochondria and microscopy.

Moreover key data need to be complemented. What is the membrane fluidity in the plasma membrane versus the fluidity in the inner and outer mitochondrial membrane? The authors should provide more data to test if the phenotypes are related to defective plasma membranes or defective mitochondrial membranes.

For HUVEC cells the authors found no changes in the FRAP assay with palmitate and KD but fail to comment on this. Why not do both assays for both cell types and explain the differences? Also, it would be good to examine a phenotype different than these membrane fluidity surrogate assays, such as cell death or UPR activation.

3) Only a small spectrum of lipids is analyzed in this study and lipid data in Figure 4 present the different acyl chains of PE lipids. However, the molar abundance of PE lipids itself is an important regulator of membrane fluidity in cells. PE is not the most abundant lipid and other lipids could contribute substantially to the observed phenotypes. The authors should provide more evidence related to the other phospholipids (e.g. What is the abundance of PC versus PE?). Most importantly, the authors should test the acyl chain composition in cardiolipin, which is specific to mitochondria.

4) Alternative models, which might explain a contribution of mitochondrial acyl-CoA synthetases to the cellular resistance against lipo toxicity, such as a general lack of electron donors for fatty acid desaturation due to mitochondrial failure caused by palmitate (in mammalian cells) or by the *paqr-*2 and *iglr-2* mutants, were not addressed. What are the functional parameters of mitochondria and the respiratory chain during SFA-mediated stress? What is the role of the identified Acyl-CoA synthetases under these conditions? The authors should exclude alternative models more rigorously, or minimally edit their Discussion to provide a more balanced view of the alternatives.

5) It would be helpful to add flow chart of diagram in the supplement illustrating epistasis in the first paragraph of the Results, with which genes/phenotypes *acs-13* enhances vs. suppresses vs. no effect. This is very interesting, but difficult to follow and would be easier to understand with a diagram. A final model would also be helpful in the final figure.

6) Does *acs-13* expression change in any of the published data sets for *mdt-15, nhr-49* or other genes in these pathways?

---

## [Author Response]

Specific points raised during the review are:1) The authors provide no data to test their primary hypothesis that acs-13 causes increased oxidation of PUFAs. This should be addressed. The preference of mitochondrial acyl-CoA synthetases for PUFAs is crucial to the presented model, but has not been directly addressed. The authors should provide evidence for a selectivity of identified Acyl-CoA synthetases. Alternatively, they should test if Acyl-CoA synthetases with a clearly defined selectivity can phenocopy the functions of the enzymes identified in this study, when targeted to mitochondria.2) Related to this, the subcellular localization of ACSLs is central to the presented model. The microscopic evidence for a mitochondrial localization, however, is not particularly strong. The authors should validate the subcellular localization by alternative means.Multiple reviewers noted that the mammalian data need to be strengthened. Related to this, the membrane fluidity assays are poorly explained in this paper. The localization of the mammalian ACSL1 could be studied by complementary methods: biochemical isolation/enrichment of mitochondria and microscopy.Moreover key data need to be complemented. What is the membrane fluidity in the plasma membrane versus the fluidity in the inner and outer mitochondrial membrane? The authors should provide more data to test if the phenotypes are related to defective plasma membranes or defective mitochondrial membranes.For HUVEC cells the authors found no changes in the FRAP assay with palmitate and KD but fail to comment on this. Why not do both assays for both cell types and explain the differences? Also, it would be good to examine a phenotype different than these membrane fluidity surrogate assays, such as cell death or UPR activation.

Both points concerned important weaknesses in our original manuscript, namely asking us to provide stronger data for the mitochondrial association of ACS-13/ACSL1, as well as documenting better the consequences of ACS-13/ACSL1 mutation/depletion. We are actually very thankful to the reviewers for essentially asking us to be more rigorous about having experimental evidence for each of our key assertions/interpretations. We now provide a wealth of new data addressing these points, and have adjusted our interpretations to not exceed the experimental evidence. Specifically:

1) The localization of ACS-13 to mitochondria in *C. elegans* is now much better supported. Most importantly, we now included Western blots showing that a GFP-tagged ACS-13 protein is enriched in mitochondria fraction (Figure 2—figure supplement 1B-C). Also, we now show that the in vivo ACS-13::GFP reporter does not co-localize with the ER marker mCherry::SP12 (Figure 2—figure supplement 1A). [This is presented in the subsection “ACS-13 is localized to mitochondria of intestinal and hypodermal cells”.]

2) Also in *C. elegans* we now show that the *acs-13* mutant exhibits mitochondria morphology defects (about 50% of mutant worms show abnormal mitochondria; Figure 2—figure supplement 1D-E). This new data supports the idea that ACS-13 has a mitochondria-related function. [This is presented in the subsection “ACS-13 is localized to mitochondria of intestinal and hypodermal cells”.]

3) The localization of ACSL1 to mitochondria in human HEK293 cells is also now supported by Western blot on purified subcellular fractions (Figure 7B; the entire Figure 7 is new). [This is presented in the subsection “.ACSL1 silencing causes changes in lipid composition and mitochondria homeostasis”]

4) Membrane fluidity is now scored in HEK293 cells using both the FRAP and Laurdan dye methods (which are now both briefly explained in the Results section). Both methods show that ACSL1 siRNA protects against rigidification by palmitate. The Laurdan dye method also provides some spatial resolution and indicated improved membrane fluidity across the cell when ACSL1 is knocked down, which we now point out in the manuscript. [This is presented in the subsection “ACSL1 knockdown promotes membrane fluidity in the presence of palmitate in human cells”.]

5) Several additional cellular phenotypes were examined. In particular, ACSL1 siRNA dampened the UPR activation that occurs when membranes are challenged by rigidification, which again supports the idea that ACSL1 silencing is beneficial in this context. This was scored by testing for UPR response gene expression using qPCR (Figure 5—figure supplement 2A). ACSL1 siRNA did not cause a change in viability based on a Trypan blue assay (Figure 5—figure supplement 2B), nor in the expression of other ACSLs (Figure 5—figure supplement 2C). Finally, ACSL1 levels themselves were no changed by silencing ADIPOR2 or TLCD1/TLCD2 (Figure 5—figure supplement 2F), which we showed can also act as suppressors of ADIPOR2 siRNA effects in a previous paper (Ruiz et al., 2018). ACSL1 therefore is not downstream of either ADIPOR2 or the TLCDs. [This is presented in the subsection “ACSL1 knockdown promotes membrane fluidity in the presence of palmitate in human cells”.]

6) We now provide Seahorse data indicating that ACSL1 siRNA does not significantly impair mitochondria respiration. (Figure 7H-I and Figure 7—figure supplement 1D-H). [This is presented in the subsection “ACSL1 silencing causes changes in lipid composition and mitochondria homeostasis”.] This is consistent with the lack of strong phenotypes in the *C. elegans acs-13* single mutant (which did show mitochondria morphology defects but no growth/lifespan defects) [This is discussed in the third paragraph of the Discussion]. Additional mitochondrial defects were also documented in the new lipidomics data (see answer to point 3, below).

7) In describing the HUVEC results, we now indicate that ACSL1 knockdown has no effect on basal condition because there is no membrane homeostasis challenge (presumably membrane homeostasis is robust enough under basal conditions). *[This is mentioned in the last paragraph before the Discussion.]*

Considering the wealth of new data on mitochondria localization and mitochondria lipid defects, we are confident in our conclusion that ACSL1 depletion in *C. elegans* and human cells results in mitochondrial defects due to abnormal LCFA import. Our revised model, i.e. that mitochondria defects explain the PE and PC changes in FA content, does not specifically require changes in substrate specificity for beta-oxidation, which was not measured for the revised manuscript. We did however measure mitochondrial acylcarnitine content and found that ACSL1 silencing caused a large increase in 16:0 species, likely at the expense of other LCFA species. Additionally, several previous studies (e.g. Grevengoed et al., 2015; Kuwala et al., 2014) have documented a preference by ACSL1 for 18:2 and other LCPUFAs, though again this is not critical for our revised model. [Please consider the entirety of the new results and the revised discussion and model in Figure 9.]

3) Only a small spectrum of lipids is analyzed in this study and lipid data in Figure 4 present the different acyl chains of PE lipids. However, the molar abundance of PE lipids itself is an important regulator of membrane fluidity in cells. PE is not the most abundant lipid and other lipids could contribute substantially to the observed phenotypes. The authors should provide more evidence related to the other phospholipids (e.g. What is the abundance of PC versus PE?). Most importantly, the authors should test the acyl chain composition in cardiolipin, which is specific to mitochondria.

This point concerned the need to analyze more lipid types than in our original manuscript. We have now done exhaustive new analyses of several lipid classes. This required setting up new collaborations and analysis pipelines. The following new lipids/parameters were analyzed: lysophosphatidylcholine FA composition, TAG FA composition, PC/PE ratio, free cholesterol/PC ratio, ceramides, dihydroxyceramides, glucosylceramides, sphingomyelins and lactosylceramides levels and FA composition, π and PS were all analyzed both in presence or absence of palmitate. The levels and composition of mitochondrial cardiolipins and the levels of palmitoylcarnitine were also determined. The new data is presented in Figures 6B-E, Figure 6—figure supplements 1 and 2 and Figure 7C-F and Figure 7—figure supplement 1C. The main conclusions are that ACSL1 siRNA results in decreased cardiolipin content in mitochondria that is accompanied by increased PUFAs in most cellular membrane phospholipids (especially PCs, PEs, and LPCs), but does not cause changes in TAGs, PC/PE ratio or cholesterol/PC ratios. This, and extensive cited literature on ACSL1, again agrees with the interpretation that ACSL1 is important for channeling LCFAs to mitochondria: its absence results in abnormal mitochondrial lipid composition. [This is presented in the subsection “ACSL1 silencing causes changes in lipid composition and mitochondria homeostasis”.]

4) Alternative models, which might explain a contribution of mitochondrial acyl-CoA synthetases to the cellular resistance against lipo toxicity, such as a general lack of electron donors for fatty acid desaturation due to mitochondrial failure caused by palmitate (in mammalian cells) or by the paqr-2 and iglr-2 mutants, were not addressed. What are the functional parameters of mitochondria and the respiratory chain during SFA-mediated stress? What is the role of the identified Acyl-CoA synthetases under these conditions? The authors should exclude alternative models more rigorously, or minimally edit their Discussion to provide a more balanced view of the alternatives.

We have now made several changes in the Discussion and revised our model so that it is less speculative and stays within the bounds of our data. In particular, the revised model in Figure 9 now unifies the *C. elegans* and human cell findings into a simpler concept whereby ACSL1/ACS-13 contributed to LCFA utilization (without specifying whether the fate is mostly beta-oxidation or use for mitochondria membrane homeostasis). Absence of ACSL1/ACS-13 leads to fewer/different LCFAs being channeled to mitochondria, causing the observed mitochondria defects but also allowing more LCFAs, many of which are PUFAs, to be incorporated into phospholipids such as PCs and PEs. The Discussion has also been adjusted to provide a more balanced view of our data. [This is presented in the subsection “ACSL1 silencing causes changes in lipid composition and mitochondria homeostasis” and discussed throughout the Discussion.]

We also suggest alternative hypothesis in a new paragraph, specifically mentioning possible roles for ACSL1 in channeling LCFAs from/to TAGs or peroxisomes. [This is presented in the fourth paragraph of the Discussion.]

5) It would be helpful to add flow chart of diagram in the supplement illustrating epistasis in the first paragraph of the Results, with which genes/phenotypes acs-13 enhances vs. suppresses vs. no effect. This is very interesting, but difficult to follow and would be easier to understand with a diagram. A final model would also be helpful in the final figure.

The genetic interaction studies in *C. elegans* are an important strength of this manuscript and we thank the reviewers for emphasizing the need for clarity in their presentation. A diagram summarizing the previously known epigenetic interactions is now included (Figure 1—figure supplement 1), and an updated version based on the present manuscript is presented as a final figure (Figure 9—figure supplement 1). [This is presented in the second paragraph of the Introduction and in the fifth paragraph of the Discussion.]

6) Does acs-13 expression change in any of the published data sets for mdt-15, nhr-49 or other genes in these pathways?

We have now measured the expression of ACS13 in *paqr-2* and *mdt-15* mutants (it is decreased in the *mdt-15* gain-of-function mutant; Figure 2—figure supplement 1C). [This is presented in the subsection “ACS-13 is localized to mitochondria of intestinal and hypodermal cells”.] Similarly, we now also measured the expression of ACSL1 in cells where ADIPOR2 or TLCD1/2 have been knocked down (it is unchanged; Figure 5—figure supplement 2F). [This is presented in the subsection “ACSL1 knockdown promotes membrane fluidity in the presence of palmitate in human cells”.]